# RE-WEIGHTING BASED GROUP FAIRNESS REGULAR-
# IZATION VIA CLASSWISE ROBUST OPTIMIZATION

**Sangwon Jung**[1*] **Taeeon Park**[1*] **Sanghyuk Chun**[2] **Taesup Moon**[1,3†]

[1] Department of Electrical and Computer Engineering, Seoul National University
[2] NAVER AI Lab    [3] ASRI/INMC/IPAI/AIIS, Seoul National University
{s.jung,pte1235,tsmoon}@snu.ac.kr, sanghyuk.c@navercorp.com

## ABSTRACT

Many existing group fairness-aware training methods aim to achieve the group fairness by either re-weighting underrepresented groups based on certain rules or using weakly approximated surrogates for the fairness metrics in the objective as regularization terms. Although each of the learning schemes has its own strength in terms of applicability or performance, respectively, it is difficult for any method in the either category to be considered as a gold standard since their successful performances are typically limited to specific cases. To that end, we propose a principled method, dubbed as FairDRO, which unifies the two learning schemes by incorporating a well-justified group fairness metric into the training objective using a *classwise* distributionally robust optimization (DRO) framework. We then develop an iterative optimization algorithm that minimizes the resulting objective by automatically producing the correct re-weights for each group. Our experiments show that FairDRO is scalable and easily adaptable to diverse applications, and consistently achieves the state-of-the-art performance on several benchmark datasets in terms of the accuracy-fairness trade-off, compared to recent strong baselines.

## 1 INTRODUCTION

Machine learning algorithms are increasingly used in various decision-making applications that have societal impact; *e.g.*, crime assessment (Julia Angwin & Kirchner, 2016), credit estimation (Khandani et al., 2010), facial recognition (Buolamwini & Gebru, 2018; Wang et al., 2019), automated filtering in social media (Fan et al., 2021), AI-assisted hiring (Nguyen & Gatica-Perez, 2016) and law enforcement (Garvie, 2016). A critical issue in such applications is the potential discrepancy of model performance, *e.g.*, accuracy, across different sensitive groups (*e.g.*, race or gender) (Buolamwini & Gebru, 2018), which is easily observed in the models trained with a vanilla empirical risk minimization (ERM) (Valiant, 1984) when the training data has unwanted bias. To address such issues, the *fairness-aware* learning has recently drawn attention in the AI research community.

One of the objectives of fairness-aware learning is to achieve *group fairness*, which focuses on the statistical parity of the model prediction across sensitive groups. The so-called *in-processing* methods typically employ additional machinery for achieving the group fairness while training. Depending on the type of machinery used, recent in-processing methods can be divided into two categories (Caton & Haas, 2020): *regularization based* methods and *re-weighting* based methods. Regularization based methods incorporate fairness-promoting terms to their loss functions. They can often achieve good performance by balancing the accuracy and fairness, but be applied only to certain types of model architectures or tasks, such as DNNs (*e.g.*, MFD (Jung et al., 2021) or FairHSIC (Quadrianto et al., 2019)) or binary classification tasks (*e.g.*, Cov (Baharlouei et al., 2020)). On the other hand, re-weighting based methods are more flexible and can be applied to a wider range of models and tasks by adopting simpler strategy to give higher weights to samples from underrepresented groups. However, most of them (*e.g.*, LBC (Jiang & Nachum, 2020), RW (Kamiran & Calders, 2012), and FairBatch (Roh et al., 2020)) lack sound theoretical justifications for enforcing group fairness and may perform poorly on some benchmark datasets.

---

[*]Equal contribution.
[†]Corresponding author.

In this paper, we devise a new in-processing method, dubbed as Fairness-aware Distributionally Robust Optimization (FairDRO), which takes the advantages of both regularization and re-weighting based methods. The core of our method is to unify the two learning categories: namely, FairDRO incorporates a well-justified group fairness metric in the training objective as a regularizer, and optimizes the resulting objective function using a re-weighting based learning method. More specifically, we first present that a group fairness metric, *Difference of Conditional Accuracy* (DCA) (Berk et al., 2021), which is a natural extension of Equalized Opportunity (Hardt et al., 2016) to the multi-class, multi-group label settings, is equivalent (up to a constant) to the average (over the classes) of the roots of the *variances* of groupwise 0-1 losses. We then employ the Group DRO formulation, which uses the $\chi^2$-divergence ball including *quasi-probabilities* as the uncertainty set, for each class *separately* to convert the DCA (or variance) regularized group-balanced empirical risk minimization (ERM) to a more tractable minimax optimization. The inner maximizer in the converted optimization problem is then used as re-weights for the samples in each group, making a unified connection between the re-weighting and regularization-based fairness-aware learning methods. Lastly, we develop an efficient iterative optimization algorithm, which automatically produces the correct (sometimes even negative) re-weights during the optimization process, in a more principled way than other re-weighting based methods.

In our experiments, we empirically show that our FairDRO is scalable and easily adaptable to diverse application scenarios, including tabular (Julia Angwin & Kirchner, 2016; Dua et al., 2017), vision (Zhang et al., 2017), and natural language text (Koh et al., 2021) datasets. We compare with several representative in-processing baselines that apply either re-weighting schemes or surrogate regularizations for group fairness, and show that FairDRO consistently achieves the state-of-the-art performance on all datasets in terms of accuracy-fairness trade-off thanks to leveraging the benefits of both kinds of fairness-aware learning methods.

## 2 RELATED WORKS

### 2.1 IN-PROCESSING METHODS FOR GROUP FAIRNESS

Regularization based methods add penalty terms in their objective function for promoting the group fairness. Due to non-differentiability of desired group fairness metrics, they use weaker surrogate regularization terms; *e.g.*, Cov (Zafar et al., 2017b), Rényi (Baharlouei et al., 2020) and FairHSIC (Quadrianto et al., 2019) employ a covariance approximation, Rényi correlation (Rényi, 1959) and Hilbert Schmidt Independence Criterion (HSIC) (Gretton et al., 2005) between the group label and the model as fairness constraints, respectively. Jung et al. (2021) devised MFD that uses Maximum Mean Discrepancy (MMD) (Gretton et al., 2005) regularization for distilling knowledge from a teacher model and promoting group fairness at the same time. Although these methods can achieve high performance when hyperparameters for strengths of the regularization terms are well-tuned, they are sensitive to choices of model architectures and task settings as they employ surrogate regularization terms; see Sec. 5 and Appendix C.2 for more details.

Meanwhile, some other works (Agarwal et al., 2018; Cotter et al., 2019) used an equivalent constrained optimization framework to enforce group fairness instead of using the regularization formulation. Namely, they consider a minimax problem of the Lagrangian function from the given constrained optimization problem and seek to find a saddle point through the alternative updates of model parameters and the Lagrangian variables. By doing so, they can successfully control the degree of fairness while maximizing the accuracy. However, their alternating optimization algorithms require severe computational costs due to the repetitive full training of the model. Furthermore, we empirically observed that they fail to find a feasible solution when applied to complex tasks in vision or NLP domains. A more detailed discussion is in Appendix B.

As alternative in-processing methods, several re-weighting based methods (Kamiran & Calders, 2012; Jiang & Nachum, 2020; Roh et al., 2020; Agarwal et al., 2018) have also been proposed to address group fairness. Kamiran & Calders (2012) proposed a re-weighting scheme (RW) based on the ratio of the number of data points per each group. Recently, Label Bias Correction (LBC) (Jiang & Nachum, 2020) and FairBatch (Roh et al., 2020) have been developed, which adaptively adjust weights and mini-batch sizes based on the average loss per group, respectively. Perhaps, the most similar to our work is Agarwal et al. (2018), which demonstrates that a Lagrangian formulation of fairness-constrained optimization problem can be reduced to a cost-sensitive classification problem

and solved through a re-weighting learning scheme with relabling of class labels. However, their method is limited to binary classification tasks, whereas our FairDRO, using the DRO framework, can be applied to more complex tasks with non-binary class labels.

## 2.2 DISTRIBUTIONALLY ROBUST OPTIMIZATION (DRO)

Distributionally robust optimization (DRO) (Ben-Tal et al., 2009) is a general robust optimization framework that was originally used for learning a model that is robust to potential test distribution shift from the training set, by optimizing for the worst-case distribution in an uncertainty set. The DRO framework is now widely adopted in other applications, including algorithmic fairness.

**DRO in algorithmic fairness** Numerous works in the algorithmic fairness literature have recently utilized the DRO frameworks in various settings. For example, a line of works (Jiang et al., 2020; Taskesen et al., 2020; Wang et al., 2021; Mandal et al., 2020) embeds the DRO frameworks into the fairness-constrained optimization problem to make a fair model robust to distribution shifts of a test dataset. Another related line of works (Wang et al., 2020; Hashimoto et al., 2018) proposed DRO formulations for achieving fairness in more challenging settings such as learning with noisy group labels or sequential learning. In spite of their effectiveness, both of above lines of works do not use DRO as a direct tool for achieving group fairness. Similarly to ours, Rezaei et al. (2020); Yurochkin et al. (2019) adopt DRO frameworks for the purpose of achieving fairness. However, the key difference is that they either require the group labels at test time or target *individual* fairness. On the other hand, our FairDRO directly aims to achieve group fairness while not requiring group labels at test time. A more detailed discussion is in Appendix B.

**DRO in other applications** Applications of DRO are actively studied in other contexts, *e.g.*, image classification (Sagawa et al., 2020), multilingual machine translation (Oren et al., 2019; Zhou et al., 2021), long-tailed classification (Samuel & Chechik, 2021) or out-of-distribution generalization (Krueger et al., 2021; Xie et al., 2020). For example, Sagawa et al. (2020) proposed Group DRO which constructs the uncertainty set of joint distributions over the class-group label pairs to withstand the spurious correlations between the class and group labels. Although these works are effective in their applications, the notion of statistical parity across groups (*i.e.,* group fairness) has not been explicitly pursued as in our FairDRO.

## 3 NOTATIONS AND FAIRNESS CRITERION

**Notations** We consider a classification problem where each sample consists of an input $x \in \mathcal{X}$, a class label $y \in \mathcal{Y}$ and a group label $a \in \mathcal{A}$. The group label is defined by sensitive attributes such as *gender* or *race*. Given a training dataset $\mathcal{D}$ with $N$ samples (*i.e.*, $\mathcal{D} = \{(x_i, y_i, a_i)\}_{i=1}^{N}$), we aim to find a fair classifier $\theta^{\text{FAIR}} : \mathcal{X} \to \mathcal{Y}$ satisfying the given fairness criterion while maintaining the high classification accuracy. Moreover, given a loss function $\ell(\theta, (x, y)) : \Theta \times (\mathcal{X} \times \mathcal{Y}) \to \mathbb{R}$, we denote $\mathcal{L}(\theta, \mathcal{D}) \triangleq \frac{1}{|\mathcal{D}|} \sum_{i=1}^{|\mathcal{D}|} \ell(\theta, (x_i, y_i))$. Additionally, $\mathcal{D}_a$ and $\mathcal{D}_a^y$ denote the subsets of $\mathcal{D}$ that are confined to the samples with group label $a$ or class-group label pair $(y, a)$, respectively.

**Fairness criterion** The notion of fairness may depend on the different points of view on how discrimination is defined (Hardt et al., 2016; Dwork et al., 2012; Chouldechova, 2017; Berk et al., 2021). For example, one may argue that a classifier is fair if its prediction is not dependent on $a$ (*Demographic Parity*), while the other can insist that the classifier should be conditionally independent of $a$ given the true class (*Equalized Odds*). In our work, we focus on the *Equalized Conditional Accuracy* (ECA) (Berk et al., 2021), which can naturally cover the multi-class, multi-group label setting.

A classifier $\theta$ satisfies ECA if all of the accuracies among groups are the same for each given class, *i.e.*, $\forall a, a' \in \mathcal{A}, y \in \mathcal{Y}, P(\widehat{Y} = y | A = a, Y = y) = P(\widehat{Y} = y | A = a', Y = y)$, where $\widehat{Y}$ is the prediction of $\theta$. We measure the degree of unfairness with *Difference of Conditional Accuracy* (DCA), denoted as $\Delta_{\text{DCA}}$, by taking the average of the maximum accuracy gaps among groups over $y$:

$$\Delta_{\text{DCA}} \triangleq \frac{1}{|\mathcal{Y}|} \sum_{y \in \mathcal{Y}} \Delta_y, \ \Delta_y \triangleq \max_{a, a'} |P(\widehat{Y} = y | A = a, Y = y) - P(\widehat{Y} = y | A = a', Y = y)|. \quad (1)$$

*Remark:* We note that ECA has close connections with the two popular group fairness criteria, *Equalized Odds* (EO) and *Equal Opportunity* (EOpp) (Hardt et al., 2016). EO requires the conditional independence between $\widehat{Y}$ and $A$ given $Y$, and hence, we can easily observe that ECA is *equivalent* to EO for the binary classification setting. Furthermore, we argue that ECA is a natural extension of EOpp to the multi-class setting; namely, ECA requires equality of TPR among groups for each class label, which resembles the argument for justifying EOpp.

## 4 FAIRNESS-AWARE DRO (FAIRDRO)

Our goal is to unify the regularization and re-weighting based methods by incorporating the DCA (1) in the objective function for learning. To that end, we first show in Sec. 4.1 that the empirical DCA is equivalent to the average (over the classes) of the roots of the *variances* of groupwise 0-1 losses. Then, we utilize the previous result (Xie et al., 2020) given in Sec. 4.2 to make the connection between the Group DRO with the uncertainty set of $\chi^2$-divergence ball (including quasi-probabilities) and the variance regularized group-balanced empirical loss minimization. Subsequently, in Sec. 4.3, we present our training objective that applies Group DRO *separately* for each class $y$, which essentially converts the DCA (or variance) regularized group-class balanced ERM into a more tractable minimax optimization problem. Finally, Sec. 4.4 proposes an efficient iterative algorithm that does alternating minimization-maximization of our training objective.

### 4.1 EQUIVALENCE BETWEEN DCA AND VARIANCE OF GROUPWISE LOSSES

**Proposition 1** *Assume* $\ell(\boldsymbol{\theta}) = \mathbb{1}\{\boldsymbol{\theta}(\boldsymbol{x}) \neq y\}$. *Then, the following inequalities hold for all* $y \in \mathcal{Y}$,

$$\sqrt{2 \operatorname{Var}\left(\{\mathcal{L}(\boldsymbol{\theta}, \mathcal{D}_a^y)\}_{a \in \mathcal{A}}\right)} \leq \widehat{\Delta}_y \leq \sqrt{2|\mathcal{A}|^2 \operatorname{Var}\left(\{\mathcal{L}(\boldsymbol{\theta}, \mathcal{D}_a^y)\}_{a \in \mathcal{A}}\right)}, \tag{2}$$

*in which* $\widehat{\Delta}_y$ *is the empirical version of* $\Delta_y$ *and* $\operatorname{Var}(\{x_i\}_{i=1}^d) \triangleq \frac{1}{d} \sum_i (x_i - \frac{1}{d} \sum_i x_i)^2$ *is the variance of* $d$ *numbers* $\{x_i\}_{i=1}^d$. *Thus,* $\widehat{\Delta}_y$ *of a classifier is 0 if and only if* $\operatorname{Var}\left(\{\mathcal{L}(\boldsymbol{\theta}, \mathcal{D}_a^y)\}_{a \in \mathcal{A}}\right) = 0$.

The proof is in Appendix A. By taking the average over $y$ in (2), Proposition 1 implies that the empirical DCA is equivalent to the average (over $y$) of the square roots of the variances of the groupwise 0-1 losses, up to a constant. Building this equivalence, we are now able to utilize the Group DRO formulation given in the next subsection to make a connection between DCA regularization and robust optimization.

### 4.2 PRIOR WORK ON VARIANCE REGULARIZATION VIA GROUP DRO

While ERM merely minimizes the empirical loss, DRO optimizes for the worst-case distributions in an uncertainty set, as mentioned in Sec. 2.2. Since general DRO (Duchi et al., 2019) may lead to a pessimistic model that optimizes for implausible worst-case distributions, Sagawa et al. (2020) recently proposed the *Group* DRO framework, which performs at a group level and can incorporate prior knowledge about the groups[1]; it aims to obtain

$$\boldsymbol{\theta}^{\mathrm{GDRO}} \triangleq \arg \min_{\boldsymbol{\theta}} \max_{\boldsymbol{q} \in \Delta^{|\mathcal{A}|}} \sum_{a \in \mathcal{A}} q_a \mathcal{L}(\boldsymbol{\theta}, \mathcal{D}_a), \tag{3}$$

in which $\Delta^{|\mathcal{A}|}$ is the $|\mathcal{A}|$-simplex.

The applications of Group DRO are actively studied in various contexts, *e.g.*, image classification (Sagawa et al., 2020) or machine translation (Zhou et al., 2021). In particular, Xie et al. (2020) proposed Risk Variance Penalization (RVP), an OOD generalization method, using a variance regularization with respect to risks of each domain (a formulation first introduced by V-REx (Krueger et al., 2021)). The author showed the connection between the variance regularization and Group DRO, when the uncertainty set is the following $\chi^2$-divergence ball with radius $\rho > 0$:

$$\mathcal{Q}_\rho \triangleq \left\{ \boldsymbol{q} \in \mathbb{R}^{|\mathcal{A}|} : \sum_{a \in \mathcal{A}} q_a = 1, \ D_\phi\left(\boldsymbol{q} \,\|\, \frac{1}{|\mathcal{A}|}\mathbb{1}\right) \leq \rho \right\}, \tag{4}$$

---

[1]In this subsection, the term "group" is used in a more inclusive manner. Namely, the group here refers to any hierarchical structure in the data distribution, *e.g.*, domain, environment, and sensitive attribute.

in which $D_\phi(p\|q) \triangleq \sum_i q_i \phi(p_i/q_i)$ with $\phi(t) = (t-1)^2$, and $\frac{1}{|\mathcal{A}|}\mathbf{1} \in \mathbb{R}^{|\mathcal{A}|}$ denotes the uniform distribution. Note (4) does not have the nonnegativity constraint on $q$, and hence, depending on $\rho$, $\mathcal{Q}_\rho$ can also include *quasi-probabilities*, which have negative components as well. Namely, Xie et al. (2020) shows the following lemma that the inner maximization of Group DRO with $\mathcal{Q}_\rho$ uncertainty set is equivalent to the group-balanced empirical loss plus the group-wise variance term as a regularizer.

**Lemma 1** *(Xie et al., 2020, Proposition 1) For any finite loss $\ell(\boldsymbol{\theta}, (\boldsymbol{x}, y))$ and $\mathcal{Q}_\rho$, we have*

$$\max_{\boldsymbol{q} \in \mathcal{Q}_\rho} \sum_{a \in \mathcal{A}} q_a \mathcal{L}(\boldsymbol{\theta}, \mathcal{D}_a) = \frac{1}{|\mathcal{A}|} \sum_a \mathcal{L}(\boldsymbol{\theta}, \mathcal{D}_a) + \sqrt{\rho \operatorname{Var}\left(\{\mathcal{L}(\boldsymbol{\theta}, \mathcal{D}_a)\}_{a \in \mathcal{A}}\right)}. \tag{5}$$

To be self-contained, the proof of the lemma is given in Appendix A.

### 4.3 TRAINING OBJECTIVE OF FAIRDRO

To make the connection between Proposition 1 and Lemma 1 and realize the DCA regularization for fairness-aware training, we propose to apply the Group DRO *separately* for each class $y$. Namely, we define our FairDRO as minimizing the average of the worst-case losses defined for each class $y$:

$$\boldsymbol{\theta}^{\text{FairDRO}} \triangleq \arg\min_{\boldsymbol{\theta}} \frac{1}{|\mathcal{Y}|} \sum_{y \in \mathcal{Y}} \max_{\boldsymbol{q}^y \in \mathcal{Q}_\rho} \sum_{a \in \mathcal{A}} q_a^y \mathcal{L}(\boldsymbol{\theta}, \mathcal{D}_a^y), \tag{6}$$

in which $\mathcal{Q}_\rho$ is as defined in (4), $\boldsymbol{q}^y = \{q_a^y\}_{a \in \mathcal{A}}$ is a quasi-probability vector, and $\rho$ is a tunable hyperparameter. We stress that while applying Group DRO by treating each pair $(y, a)$ as a "group" has been considered in Sagawa et al. (2020), applying Group DRO separately for each class as in FairDRO has not been considered before. Now, we present our key result which follows by combining Proposition 1 and Lemma 1.

**Corollary 1** *Assume $\ell(\boldsymbol{\theta}) = \mathbb{1}\{\boldsymbol{\theta}(\boldsymbol{x}) \neq y\}$. Then, given a dataset $\mathcal{D}$, there is a corresponding positive constant $C_\mathcal{D}$ in the range of $[\sqrt{2}, \sqrt{2}|\mathcal{A}|]$ such that $\boldsymbol{\theta}^{\text{FairDRO}}$ also achieves*

$$\min_{\boldsymbol{\theta}} \left\{ \frac{1}{|\mathcal{Y}||\mathcal{A}|} \sum_{(y,a)} \mathcal{L}(\boldsymbol{\theta}, \mathcal{D}_a^y) + C_\mathcal{D} \sqrt{\rho} \widehat{\Delta}_{DCA} \right\}. \tag{7}$$

The corollary implies that when the 0-1 loss is used, FairDRO is a *regularization based method*, as described in Sec. 2.1, with the *exact* DCA serving as a regularizer. We emphasize that we do not solve the minimization in (7) directly. $\widehat{\Delta}_{\text{DCA}}$ can have high variance when estimated from a small mini-batch, particularly when the number of groups is large, and is even hard to be minimized due to non-differentiability. We indeed empirically observe in Sec. 5.4 that directly solving (7) using approximation of $\widehat{\Delta}_{\text{DCA}}$ with a differentiable cross-entropy loss performs poorly. Instead, we solve a much more tractable minimax optimization in (6) where the maximization is a simple linear optimization and the minimization is for the *re-weighted* group losses.

At this point, it is now clear that during solving (6), FairDRO produces the re-weights $\{q_a^y\}$'s for each $(y, a)$, which makes it also categorized as a *re-weighting based method*. The following proposition shows how the re-weights are characterized and controlled by the hyperparameter $\rho$.

**Proposition 2** *For $\mathcal{Q}_\rho$ in (4) with $\rho > 0$, each element of the optimum $\boldsymbol{q}^{y*}$ for the inner-maximization of (6) can be obtained by a closed-form solution:*

$$q_a^{y*} = \frac{1}{|\mathcal{A}|} + \sqrt{\frac{\rho}{|\mathcal{A}|}} \times \frac{\mathcal{L}(\boldsymbol{\theta}, \mathcal{D}_a^y) - \sum_{a \in \mathcal{A}} \frac{1}{|\mathcal{A}|} \mathcal{L}(\boldsymbol{\theta}, \mathcal{D}_a^y)}{\|\mathcal{L}(\boldsymbol{\theta}, \mathcal{D}_a^y) - \sum_{a \in \mathcal{A}} \frac{1}{|\mathcal{A}|} \mathcal{L}(\boldsymbol{\theta}, \mathcal{D}_a^y)\|_2}, \tag{8}$$

*and is in the range of $\left[ \frac{1}{|\mathcal{A}|} - \frac{\sqrt{\rho(|\mathcal{A}|-1)}}{|\mathcal{A}|}, \frac{1}{|\mathcal{A}|} + \frac{\sqrt{\rho(|\mathcal{A}|-1)}}{|\mathcal{A}|} \right]$.*

The proof is in Appendix A. Proposition 2 shows that some elements of $\boldsymbol{q}^{y*}$ can be negative depending on the radius $\rho$ and the groupwise loss $\mathcal{L}(\boldsymbol{\theta}, \mathcal{D}_a^y)$. Moreover, it shows that the range of each element of the optimum $\boldsymbol{q}^{y*}$ is automatically determined by a radius $\rho$ and the number of groups $|\mathcal{A}|$, which

means FairDRO controls the degree of the group penalization by varying $\rho$. We will show in Sec. 5 that we can find a proper degree of penalization by tuning $\rho$ so that the performances on various datasets in terms of accuracy-fairness trade-off are significantly improved.

A crucial difference from typical re-weighting based methods is that our re-weights are obtained as the solution of an optimization problem as opposed to those in Kamiran & Calders (2012); Jiang & Nachum (2020) that are heuristically obtained based on the ratio of the number of data points or losses for each group. Also, we allow negative values for $q_a^y$ in $\mathcal{Q}_\rho$, which enables penalizing the high accuracy group more aggressively than typical re-weighting methods that only control the *positive* weights for each group $a$ and class $y$.

### 4.4 AN EFFICIENT ITERATIVE OPTIMIZATION FOR FAIRDRO

A canonical way of solving the minimax optimization in DRO is to use a primal-dual method (Nemirovski et al., 2009) in which one alternates between a regular gradient descent on $\boldsymbol{\theta}$ and an exponentiated gradient (EG) ascent (Kivinen & Warmuth, 1997) on $\boldsymbol{q}$. However, the EG algorithm is widely used when the variables lie in the probability simplex because it is a special case of mirror descent when the convex function for the Bregman divergence is the negative Shannon entropy. Thus, EG cannot be applied to solving (6) since our uncertainty set also allows the quasi-probability for $\boldsymbol{q}^y$.

Hence, we employ the *smoothed* Iterated Best Response (IBR) which is a variant of IBR recently used for solving a DRO problem in (Zhou et al., 2021). Using standard IBR for the inner maximization step, $\boldsymbol{q}$ is updated as the closed-form solution of the maximization (*i.e.*, (8)), instead of the gradient ascent of $\boldsymbol{q}$. It is known that solving the minimax optimization using the standard IBR has a convergence guarantee w.r.t $\boldsymbol{\theta}$ under some regularity assumptions (Zhou et al., 2021). However, it does not imply the convergence of $\boldsymbol{q}^y$, and we empirically observed in Appendix D.6 that the $\boldsymbol{q}^{y*,t}$ oscillates unsteadily when the loss for each group fluctuates. To that end, we applied the learning rate scheduling of $\eta_{\boldsymbol{q}}^t = 1 - \frac{t}{T}$ with $T$ being the number of total epochs, which results in updating $\boldsymbol{q}^{y,t}$ *smoothly*. Namely, our overall update rules for $\boldsymbol{\theta}$ and $\boldsymbol{q}^y$ are

$$\boldsymbol{\theta}^{t+1} \leftarrow \boldsymbol{\theta}^t - \eta_{\boldsymbol{\theta}}^t \nabla_{\boldsymbol{\theta}} \Big( \frac{1}{|\mathcal{Y}|} \sum_{y \in \mathcal{Y}} \sum_{a \in \mathcal{A}} q_a^{y,t} \mathcal{L}(\boldsymbol{\theta}^t, \mathcal{D}_a^y) \Big), \quad \text{(update } \boldsymbol{\theta}) \tag{9}$$

$$\boldsymbol{q}^{y,t+1} \leftarrow (1 - \eta_{\boldsymbol{q}}^t) \boldsymbol{q}^{y,t} + \eta_{\boldsymbol{q}}^t \boldsymbol{q}^{y*,t}, \quad \forall y \in \mathcal{Y}, \quad \text{(update } \boldsymbol{q}^y) \tag{10}$$

$$\text{in which} \quad \boldsymbol{q}^{y*,t} \triangleq \underset{\boldsymbol{q}^y \in \mathcal{Q}_\rho}{\arg\max} \sum_{a \in \mathcal{A}} q_a^y \mathcal{L}(\boldsymbol{\theta}^{t+1}, \mathcal{D}_a^y),$$

---

**Algorithm 1:** FairDRO Iterative Optimization

**Input** : Dataset $\{\mathcal{D}_a^y\}_{y \in \mathcal{Y}, a \in \mathcal{A}}$, radius $\rho$, epoch $T$, iteration $I$, mini-batch size $B$

1 Randomly initialize $\boldsymbol{\theta}^0$
2 Set $\boldsymbol{q}^{y,0} \leftarrow \frac{1}{|\mathcal{A}|}\mathbb{1}$ for $y \in \mathcal{Y}$, in which $\mathbb{1} \in \mathbb{R}^{|\mathcal{A}|}$
3 **for** $t = 0$ **to** $T - 1$ **do**
4     // $\boldsymbol{\theta}$-update
5     **for** $i = 0$ **to** $I - 1$ **do**
6         Sample a mini-batch $\{(\boldsymbol{x}_j, y_j, a_j)\}_{j \in [B]}$, equally from $\mathcal{D}_a^y$ for each $(y, a) \in \mathcal{Y} \times \mathcal{A}$.
7         Update $\boldsymbol{\theta}^t$ with (9) using the mini-batch
8     **end**
9     // $\boldsymbol{q}$-update
10     Update $\boldsymbol{q}^{y,t}$ for each $y \in \mathcal{Y}$ with (10)
11 **end**

**Output :** $\boldsymbol{\theta}^T, \{\boldsymbol{q}^{y,T}\}_{y \in \mathcal{Y}}$

---

and $\eta_{\boldsymbol{\theta}}^t$ and $\eta_{\boldsymbol{q}}^t$ denote the learning rates for $\boldsymbol{\theta}$ and $\boldsymbol{q}^y$, respectively. By applying the smoothed IBR in (10), we empirically observe improvements in the group fairness and more stable convergence of $\boldsymbol{q}^y$ (detailed results are given in Appendix D.6). We note that for (7) to hold, we use 0-1 loss in (6), which makes our training objective still non-differentiable with respect to $\boldsymbol{\theta}$. Thus, in practice, we solves the cross-entropy loss for the outer minimization (9) and the 0-1 loss for the inner maximization (10). We also note that $\boldsymbol{\theta}$ is updated in a mini-batch manner, whereas $\boldsymbol{q}$ is updated after computing the group losses for all training data points. Algorithm 1 illustrates the summary of FairDRO optimization scheme.

## 5 EXPERIMENTAL RESULTS

In this section, we demonstrate the versatility of FairDRO by comparing it with various baseline methods on three modalities of datasets: (1) two tabular datasets, UCI Adult (Dua et al., 2017) and ProPublica COMPAS (Julia Angwin & Kirchner, 2016), (2) two vision datasets, UTKFace (Zhang et al., 2017), CelebA (Liu et al., 2015) (in Appendix D.2) and (3) a natural language dataset,

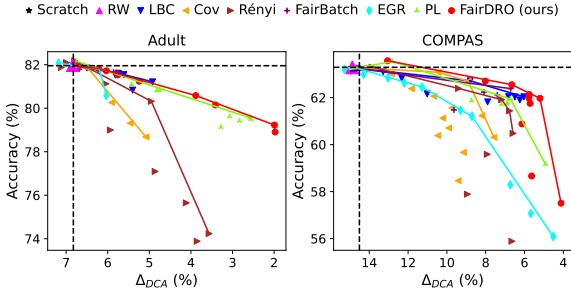

Figure 1: **The trade-offs between accuracy and DCA on Adult (left) and COMPAS (right)**. The performances of "Scratch" are highlighted by the black dotted lines. Each point is the result corresponding to a different hyperparameter of each method. Each line is a convex envelop of points of the corresponding method.

CivilComments-WILDS (Koh et al., 2021). We use logistic regression, ResNet18 (He et al., 2016), and a pre-trained BERT (Devlin et al., 2019; Wolf et al., 2019) as the classifier for each modality, respectively. For the evaluation, we plotted the convex envelops of the trade-offs between the accuracy and $\Delta_{\mathrm{DCA}}$ for varying hyperparameters, *i.e.*, the pareto frontiers. As a single number evaluation metric, we also reported the best DCA with at least 95% accuracy of the vanilla trained model (Scratch) in the Appendix D.1. Since the most datasets are severely skewed toward a specific group or class, we report the balanced accuracy over all combinations of groups and classes as Bellamy et al. (2018) (the issue discussed more in Appendix D.7). Moreover, we give a thorough ablation study on our design choice in (6), namely, the classwise DRO formulation and the uncertainty set (4) in Sec. 5.4 and Appendix D.4, respectively. We further visualize the re-weights produced by FairDRO in Sec. 5.4. More implementation details are given in Appendix C.

**Comparison methods**. We compare FairDRO with vanilla training without any fairness constraint (Scratch) and nine in-processing fair-training methods; three re-weighting based methods (RW (Kamiran & Calders, 2012), LBC (Jiang & Nachum, 2020), FairBatch (Roh et al., 2020)), four regularization based methods (Cov (Zafar et al., 2017b), FairHSIC (Quadrianto et al., 2019), Rényi (Baharlouei et al., 2020), MFD (Jung et al., 2021)) and two methods employing a constrained optimization (EGR (Agarwal et al., 2018), PL (Cotter et al., 2019)). Since we consider a group-class balanced accuracy, we modified the objective functions of all baselines including Scratch such that they use the group-class balanced ERM loss, not the standard ERM loss.[2] We note that Cov, EGR and Rényi are task-dependent, and FairHSIC and MFD are model-dependent; namely, Cov and EGR are designed only for binary classification tasks, Rényi is for tasks with binary groups, and FairHSIC and MFD are for DNN-based classifiers. We also note that all baselines except RW were implemented for targeting EO. Considering that ECA is equivalent to EO only in binary classification as we mentioned in Section 3, we also reported the difference of EO (DEO) for multi-class classification tasks in Appendix D.1. We provide more details of implementations of the baselines in Appendix C.

## 5.1 TABULAR DATASETS

Two tabular datasets, UCI Adult (Dua et al., 2017) (Adult) and ProPublica COMPAS (Julia Angwin & Kirchner, 2016) (COMPAS), are used for the benchmark. Both tasks are binary classification tasks (predicting whether the income of an individual exceeds $50K per a year and whether a defendant re-offends, respectively) with binary sensitive groups ("Female" and "Male", and "Caucasian" and "Non-Caucasin", respectively). Following pre-processing of Bellamy et al. (2018), each dataset includes 45K and 5K rows, respectively.

Fig. 1 shows the accuracy-fairness trade-offs for varying hyperparameters on Adult and COMPAS. We note that there is only a single score for RW because it has no controllable hyperparameter. In the figure, we observe that FairDRO is placed to the most top-right area among methods, showing the best trade-off, and even achieves the lowest level of DCA with a slight loss of accuracy. This implies that FairDRO successfully finds better weights (*i.e.*, quasi-probability $q^y$) than other re-weighting based methods and employs a better regularizer than other regularization based methods. Furthermore, our FairDRO can provide pareto-frontiers in a wider range than other re-weighting baselines by introducing negative weights.

---

[2]The results of employing original objective functions are reported in Appendix D.3. From the results, we observe that using the balanced ERM loss improves the balanced accuracy and DCA for the most methods.

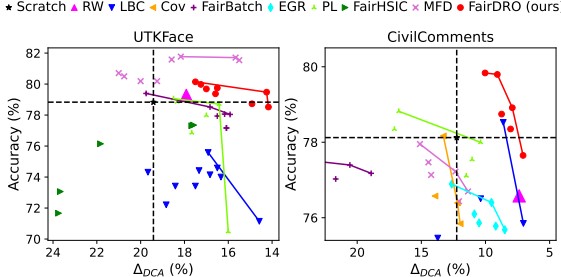

Figure 2: **The trade-offs between accuracy and DCA on UTKFace (left) and CivilComments (right).** Details are the same as Fig. 1.

## 5.2 VISION DATASETS

We also evaluate FairDRO on UTKFace (Zhang et al., 2017), a face dataset with multi-class and multi-group labels. UTKFace includes more than 20K images with age, gender and ethnicity labels. We divide the age labels into three classes ("0 to 19", "20 to 40", and "more than 40"), and set them as the class labels. We also use four ethnic classes (*"White", "Black", "Asian",* and *"Indian"*) as the group labels. The description and results on another vision dataset, CelebA (Liu et al., 2015), are given in Appendix D.2.

From Fig. 2 (left), we confirm that FairDRO significantly improves the trade-off, compared to most baselines. For MFD, we note that it provides a competitive pareto-frontier with higher accuracy than FairDRO, thanks to the knowledge distillation effect. However, MFD requires additional computations for training a teacher model, and its high performance is sensitive to task domains (refer to the next result on the NLP task). We again observe that the re-weighting based baselines are ineffective in achieving fairness on UTKFace, while FairDRO achieves strong performance due to its systematic optimization of the group weights. We further note that much lower accuracy of PL implies that its minimax optimization fails to produce good performance due to difficulty of finding feasible saddle points. Since EO is not equivalent to ECA for non-binary target task as aforementioned above, we additionally report the difference of EO (DEO) for UTKFace in Appendix D.1 and show FairDRO again outperforms other baselines in DEO.

## 5.3 LANGUAGE DATASET

In order to show the versatility of FairDRO, we additionally consider a natural language dataset, CivilComments-WILDS (Koh et al., 2021). CivilComments is a large collection of comments on online articles taken from the Civil Comments platform. It comprises 450K comments which are annotated with identity attributes by 10 crowdworkers and majority votes. The CivilComments task is to predict whether a given comment is toxic or not. We set ethnicity as the sensitive attribute and extract the subset of comments that contain identities w.r.t. ethnicity (*"Black", "White", "Asian", "Latino"* and *"Ohters"*). Note that the dataset has continuous values for the identities (between 0 and 1), hence, we binarized the continuous group values by setting the maximum as 1 and the others as 0, since the existing in-processing methods are only available to handle discrete group labels.

The results are shown in Fig. 2 (right). Note that since a pre-trained BERT is fine-tuned only for small epochs, the update for $q^y$ of FairDRO and weights of re-weighting baselines is executed per 100 iterations, instead of every epoch. We empirically observed that FairHSIC fails to converge due to the gradient exploding. Consistent with the previous results, FairDRO achieves the best trade-off on CivilComments. Note that although MFD is very competitive on UTKFace, it is not effective on CivilComments.

## 5.4 ANALYSIS

**Ablation study**. Fig. 3 shows the accuracy-fairness trade-off for each ablated version of FairDRO on each dataset. "FairDRO (w/o DRO)" directly solves (7) by approximating both terms in the objective with the cross-entropy loss; namely, both terms are replaced with the group-class balanced average cross-entropy loss and the average (over $y$) of the maximum gaps of groupwise cross-entropy losses. The "RVP (w /classwise)" is also similar, but uses the average (over $y$) of the *variances* of the groupwise cross-entropy losses (as shown in (5)) as a regularization term instead of the direct approximation of $\widehat{\Delta}_{\text{DCA}}$. FairDRO (w/o classwise) defines only one uncertainty set over the $(y, a)$

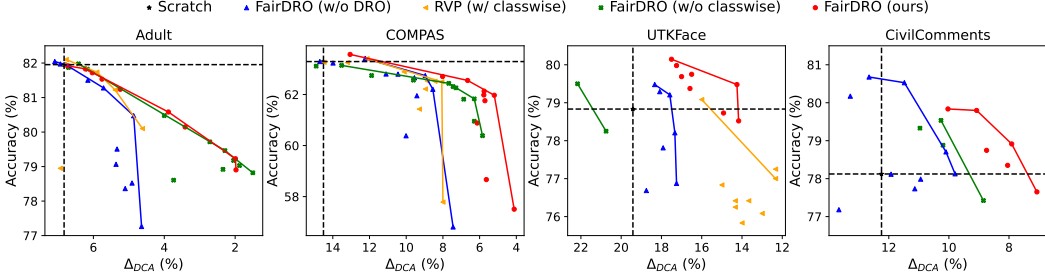

Figure 3: **Ablation study of FairDRO**. RVP and FairDRO (w/o classwise) are for the ablation study of our key two components, the DRO formulation, and classwise treatment, respectively. The best performance of each result is in Tab. D.4.

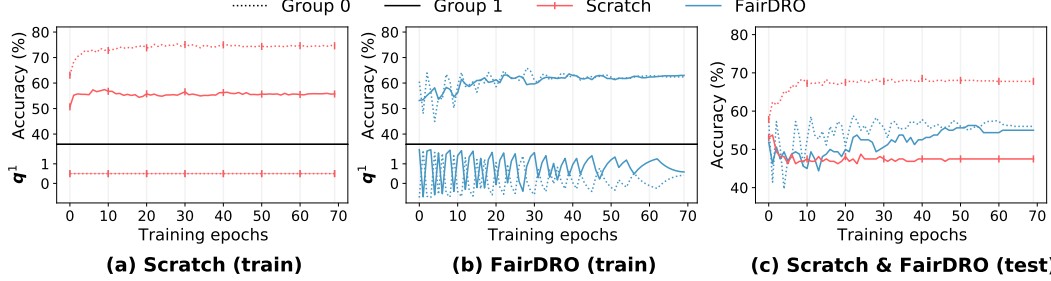

Figure 4: **Accuracies and loss weights for each group ($q$) on COMPAS**. The groupwise training accuracies and loss weights during training for Scratch (a) and FairDRO (b) are shown. Groupwise test accuracies are shown in (c). The reported accuracies and weights are only for label 1, while label 0 results are in Appendix D.5.

pair in (6) for the ablation study of the classwise treatment. In Fig. 3, we note that RVP (w/ classwise) on CivilComments fails to converge by the gradient exploding. Our observations from the figures are as follows. First, we see that FairDRO achieves consistently better pareto-frontiers on all datasets than FairDRO (w/o DRO) and RVP (w/classwise), which asserts that solving (7) by a re-weighting learning scheme via our classwise DRO formulation is essential for better performance and stable learning. Second, we clearly observe that FairDRO (w/o classwise) shows sub-optimal accuracy-DCA trade-offs (except for Adult), indicating that our classwise DRO formulation is critical for the optimal performance since it makes the connection with the DCA regularization more clearly. Further ablation studies for the choice of the uncertainty set are in Appendix D.4.

**Visualization of $q^y$.** We visualize the learning dynamics of the group weights ($q^y$), including training accuracies (Fig. 4 (a) and (b)) and test accuracies (Fig. 4 (c)) for Scratch and FairDRO on COMPAS. For simplicity, we report the result for $y = 1$ (for $y = 0$ is in Appendix D.5). In Fig. 4 (a), $q^1$ assigns the same value for each group as Scratch employs the balanced ERM loss. At the top of the plot, Scratch shows a large discrepancy in accuracy between the two groups throughout training. On the other hand, Fig. 4 (b) shows FairDRO adjusts $q^1$ by assigning low values to the high accuracy group (Group 0) and high values to the low accuracy group (Group 1). Note that $q^1$ initially fluctuates and even assigns *negative* values for the high accuracy group but eventually converges via the learning rate scheduling in (10). As a result, we clearly observe both the training and test accuracy gaps between the groups smoothly become negligible as the training continues. We believe this example clearly shows the effectiveness of introducing the quasi-probability in $\mathcal{Q}_\rho$, and our FairDRO framework in achieving small test accuracy gaps between the groups, *i.e.*, $\Delta_{\text{DCA}}$.

## 6 CONCLUDING REMARK

We proposed a theoretically well-justified in-processing method for achieving group fairness. To make a clear connection between re-weighting and regularization based methods, our FairDRO employs the classwise Group DRO framework with $\chi^2$-divergence ball including quasi-probabilities as an uncertainty set. Then, the FairDRO training objective directly incorporates DCA as a regularizer and can be effectively solved by our proposed re-weighting optimization scheme. As a result, our DRO formulation enables FairDRO to have strengths of both re-weighing based and regularization based methods. Indeed, we showed through our experiments that our FairDRO is applicable to various settings and consistently achieves superior accuracy-fairness trade-off on benchmark datasets.

## ACKNOWLEDGMENTS

This work was supported in part by the NRF grant [NRF-2021R1A2C2007884] and IITP grants [No.2021- 0-01343, No.2021-0-02068, No.2022-0-00959, No.2022-0-00113]] funded by the Korean government, and SNU-NAVER Hyperscale AI Center.

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

## APPENDIX

We provide additional materials in this document. We include the proofs for Lemma 1 and Proposition 1 and 2 in Appendix A, additional related work in Appendix B, and implementation details such as optimization, implementations of baselines, and hyperparameter search in Appendix C. Finally, we report additional experimental results in Appendix D.

## A   PROOFS

**Proof of Proposition 1**   We define $\mathbf{L}^y \in \mathbb{R}^{|\mathcal{A}| \times |\mathcal{A}|}$ for brevity:

$$\mathrm{L}^y_{i,j} \triangleq \mathcal{L}(\boldsymbol{\theta}, \mathcal{D}^y_i) - \mathcal{L}(\boldsymbol{\theta}, \mathcal{D}^y_j), 1 \leq \forall i, j \leq |\mathcal{A}|.$$

If we denote $\boldsymbol{\Lambda}^y \in \mathbb{R}^{|\mathcal{A}|^2}$ as

$$\boldsymbol{\Lambda}^y \triangleq (\mathrm{L}^y_{1,1}, ..., \mathrm{L}^y_{|\mathcal{A}|,1}, \mathrm{L}^y_{1,2}, ..., \mathrm{L}^y_{|\mathcal{A}|,2}, ..., \mathrm{L}^y_{|\mathcal{A}|,|\mathcal{A}|}),$$

then, we have the following chain of equalities:

$$
\begin{aligned}
\mathrm{Var}\left(\{\mathcal{L}(\boldsymbol{\theta}, \mathcal{D}^y_a)\}_{a \in \mathcal{A}}\right) =& \frac{1}{|\mathcal{A}|} \sum_a \left(\mathcal{L}(\boldsymbol{\theta}, \mathcal{D}^y_a)\right)^2 - \left(\frac{1}{|\mathcal{A}|} \sum_a \left(\mathcal{L}(\boldsymbol{\theta}, \mathcal{D}^y_a)\right)\right)^2 \\
=& \frac{1}{2|\mathcal{A}|^2} \sum_a \sum_{a\prime} \left(\mathcal{L}(\boldsymbol{\theta}, \mathcal{D}^y_a) - \mathcal{L}(\boldsymbol{\theta}, \mathcal{D}^y_{a\prime})\right)^2 \\
=& \frac{1}{2|\mathcal{A}|^2} \|\boldsymbol{\Lambda}^y\|_2^2.
\end{aligned}
$$

Note that as we consider $\ell$ to be zero-one loss function, we have the followings:

$$
\begin{aligned}
\widehat{\Delta}_y \triangleq& \max_{a,a\prime} \left| \frac{1}{|\mathcal{D}^y_a|} \sum_{i=1}^{|\mathcal{D}^y_a|} \mathbb{1}\{\boldsymbol{\theta}(\boldsymbol{x}_i) \neq y\} - \frac{1}{|\mathcal{D}^y_{a\prime}|} \sum_{i=1}^{|\mathcal{D}^y_{a\prime}|} \mathbb{1}\{\boldsymbol{\theta}(\boldsymbol{x}_i) \neq y\} \right| \\
=& \max_{a,a\prime} |\mathcal{L}(\boldsymbol{\theta}, \mathcal{D}^y_a) - \mathcal{L}(\boldsymbol{\theta}, \mathcal{D}^y_{a\prime})| \\
=& \|\boldsymbol{\Lambda}^y\|_\infty.
\end{aligned}
$$

Since $\|\boldsymbol{\Lambda}^y\|_\infty = \max_i |\Lambda_i^y| \leq \sqrt{\sum_i (\Lambda_i^y)^2} = \|\boldsymbol{\Lambda}^y\|_2$, we obtain the second inequality of (2). Note also that $\|\boldsymbol{\Lambda}^y\|_2 = \sqrt{\sum_i (\Lambda_i^y)^2} \leq \sqrt{\sum_i \|\boldsymbol{\Lambda}^y\|_\infty^2} = \sqrt{|\mathcal{A}|^2 \|\boldsymbol{\Lambda}^y\|_\infty^2} = |\mathcal{A}| \|\boldsymbol{\Lambda}^y\|_\infty$. We thereby obtain the first inequality of (2).

From (2), we obtain $\widehat{\Delta}_y = 0$ if $\mathrm{Var}\left(\{\mathcal{L}(\boldsymbol{\theta}, \mathcal{D}^y_a)\}_{a \in \mathcal{A}}\right) = 0$ for all $y \in \mathcal{Y}$. Considering variance is always non-negative, we obtain the necessary condition that $\widehat{\Delta}_y = 0$ implies $\mathrm{Var}\left(\{\mathcal{L}(\boldsymbol{\theta}, \mathcal{D}^y_a)\}_{a \in \mathcal{A}}\right) = 0$ for all $y \in \mathcal{Y}$. ■

**Proof of Lemma 1**   Suppose that a model parameter $\boldsymbol{\theta}$ is given. As $\sum_{a \in \mathcal{A}} \left(q_a^y - \frac{1}{|\mathcal{A}|}\right) = 0$ and $\sum_{a \in \mathcal{A}} \frac{1}{|\mathcal{A}|} \mathcal{L}(\boldsymbol{\theta}, \mathcal{D}^y_a)$ is a constant, we can decompose $\sum_{a \in \mathcal{A}} q_a^y \mathcal{L}(\boldsymbol{\theta}, \mathcal{D}^y_a)$ into

$$
\begin{aligned}
\sum_{a \in \mathcal{A}} q_a^y \mathcal{L}(\boldsymbol{\theta}, \mathcal{D}^y_a) =& \sum_{a \in \mathcal{A}} \frac{1}{|\mathcal{A}|} \mathcal{L}(\boldsymbol{\theta}, \mathcal{D}^y_a) + \sum_{a \in \mathcal{A}} (q_a^y - \frac{1}{|\mathcal{A}|}) \mathcal{L}(\boldsymbol{\theta}, \mathcal{D}^y_a) \\
=& \sum_{a \in \mathcal{A}} \frac{1}{|\mathcal{A}|} \mathcal{L}(\boldsymbol{\theta}, \mathcal{D}^y_a) + \sum_{a \in \mathcal{A}} (q_a^y - \frac{1}{|\mathcal{A}|}) \left(\mathcal{L}(\boldsymbol{\theta}, \mathcal{D}^y_a) - \sum_{a \in \mathcal{A}} \frac{1}{|\mathcal{A}|} \mathcal{L}(\boldsymbol{\theta}, \mathcal{D}^y_a)\right).
\end{aligned}
$$

Let $\boldsymbol{v}^y \triangleq \boldsymbol{q}^y - \frac{1}{|\mathcal{A}|}\mathbf{1}$, where $\mathbf{1} \in \mathbb{R}^{|\mathcal{A}|}$. From $\sum_{a \in \mathcal{A}} q_a^y = 1$, we have $\sum_{a \in \mathcal{A}} v_a^y = 0$.

Considering $D_\phi\left(\boldsymbol{q} \,\middle\|\, \frac{1}{|\mathcal{A}|}\mathbf{1}\right) = \sum_{a \in \mathcal{A}} \frac{1}{|\mathcal{A}|}(|\mathcal{A}|q_a^y - 1)^2 = \sum_{a \in \mathcal{A}} |\mathcal{A}|(q_a^y - \frac{1}{|\mathcal{A}|})^2 = |\mathcal{A}| \sum_{a \in \mathcal{A}} (v_a^y)^2,$

the following problems are equivalent:

$$\max_{\boldsymbol{q}^y \in \mathcal{Q}_\rho} \sum_{a \in \mathcal{A}} q_a^y \mathcal{L}(\boldsymbol{\theta}, \mathcal{D}_a^y) = \max_{\boldsymbol{v}^y} \sum_{a \in \mathcal{A}} \frac{1}{|\mathcal{A}|} \mathcal{L}(\boldsymbol{\theta}, \mathcal{D}_a^y) + \sum_{a \in \mathcal{A}} v_a^y \Big( \mathcal{L}(\boldsymbol{\theta}, \mathcal{D}_a^y) - \sum_{a \in \mathcal{A}} \frac{1}{|\mathcal{A}|} \mathcal{L}(\boldsymbol{\theta}, \mathcal{D}_a^y) \Big)$$

$$\text{subject to } \sum_{a \in \mathcal{A}} v_a^y = 0, \|\boldsymbol{v}^y\|_2^2 \leq \frac{\rho}{|\mathcal{A}|}.$$

Also, we have the following inequalities:

$$\sum_{a \in \mathcal{A}} v_a^y \Big( \mathcal{L}(\boldsymbol{\theta}, \mathcal{D}_a^y) - \sum_{a \in \mathcal{A}} \frac{1}{|\mathcal{A}|} \mathcal{L}(\boldsymbol{\theta}, \mathcal{D}_a^y) \Big) \leq \sqrt{\sum_{a \in \mathcal{A}} (v_a^y)^2} \sqrt{\sum_{a \in \mathcal{A}} \Big( \mathcal{L}(\boldsymbol{\theta}, \mathcal{D}_a^y) - \sum_{a \in \mathcal{A}} \frac{1}{|\mathcal{A}|} \mathcal{L}(\boldsymbol{\theta}, \mathcal{D}_a^y) \Big)^2}$$

$$\tag{A.1}$$

$$\leq \sqrt{\frac{\rho}{|\mathcal{A}|}} \sqrt{|\mathcal{A}| \times \text{Var}\left(\{\mathcal{L}(\boldsymbol{\theta}, \mathcal{D}_a^y)\}_{a \in [\mathcal{A}]}\right)}, \tag{A.2}$$

in which (A.1) holds by the Cauchy-Schwartz inequality. The second inequality follows from

$$\text{Var}\left(\{\mathcal{L}(\boldsymbol{\theta}, \mathcal{D}_a^y)\}_{a \in \mathcal{A}}\right) = \frac{1}{|\mathcal{A}|} \sum_{a \in \mathcal{A}} \Big( \mathcal{L}(\boldsymbol{\theta}, \mathcal{D}_a^y) - \sum_{a \in \mathcal{A}} \frac{1}{|\mathcal{A}|} \mathcal{L}(\boldsymbol{\theta}, \mathcal{D}_a^y) \Big)^2,$$

$$\|\boldsymbol{v}^y\|_2^2 \leq \frac{\rho}{|\mathcal{A}|}.$$

The equality in (A.2) is attained if and only if the vector $\boldsymbol{v}^y$ satisfies: (i) $\boldsymbol{v}^y$ and the $|\mathcal{A}|$-dimensional vector whose $a$-th element is $\mathcal{L}(\boldsymbol{\theta}, \mathcal{D}_a^y) - \sum_{a \in \mathcal{A}} \frac{1}{|\mathcal{A}|} \mathcal{L}(\boldsymbol{\theta}, \mathcal{D}_a^y)$ are in the same direction; (ii) $\|\boldsymbol{v}^y\|_2^2 = \frac{\rho}{|\mathcal{A}|}$. This implies that the $a$-th element of $\boldsymbol{v}^y$ should be

$$v_a^y = \sqrt{\frac{\rho}{|\mathcal{A}|}} \times \frac{\mathcal{L}(\boldsymbol{\theta}, \mathcal{D}_a^y) - \sum_{a \in \mathcal{A}} \frac{1}{|\mathcal{A}|} \mathcal{L}(\boldsymbol{\theta}, \mathcal{D}_a^y)}{\sqrt{\sum_{a \in \mathcal{A}} \Big( \mathcal{L}(\boldsymbol{\theta}, \mathcal{D}_a^y) - \sum_{a \in \mathcal{A}} \frac{1}{|\mathcal{A}|} \mathcal{L}(\boldsymbol{\theta}, \mathcal{D}_a^y) \Big)^2}}$$

$$= \sqrt{\frac{\rho}{|\mathcal{A}|}} \times \frac{\mathcal{L}(\boldsymbol{\theta}, \mathcal{D}_a^y) - \sum_{a \in \mathcal{A}} \frac{1}{|\mathcal{A}|} \mathcal{L}(\boldsymbol{\theta}, \mathcal{D}_a^y)}{\sqrt{|\mathcal{A}| \times \text{Var}\left(\{\mathcal{L}(\boldsymbol{\theta}, \mathcal{D}_a^y)\}_{a \in \mathcal{A}}\right)}}.$$

Thus, Lemma 1 holds if and only if

$$q_a^y = \frac{1}{|\mathcal{A}|} + \sqrt{\frac{\rho}{|\mathcal{A}|}} \times \frac{\mathcal{L}(\boldsymbol{\theta}, \mathcal{D}_a^y) - \sum_{a \in \mathcal{A}} \frac{1}{|\mathcal{A}|} \mathcal{L}(\boldsymbol{\theta}, \mathcal{D}_a^y)}{\|\mathcal{L}(\boldsymbol{\theta}, \mathcal{D}_a^y) - \sum_{a \in \mathcal{A}} \frac{1}{|\mathcal{A}|} \mathcal{L}(\boldsymbol{\theta}, \mathcal{D}_a^y)\|_2},$$

for all $a \in \mathcal{A}$. ∎

**Proof of Proposition 2** From the proof of Lemma 1, the equality in (5) holds if and only if

$$q_a^y = \frac{1}{|\mathcal{A}|} + \sqrt{\frac{\rho}{|\mathcal{A}|}} \times \frac{\mathcal{L}(\boldsymbol{\theta}, \mathcal{D}_a^y) - \sum_{a \in \mathcal{A}} \frac{1}{|\mathcal{A}|} \mathcal{L}(\boldsymbol{\theta}, \mathcal{D}_a^y)}{\|\mathcal{L}(\boldsymbol{\theta}, \mathcal{D}_a^y) - \sum_{a \in \mathcal{A}} \frac{1}{|\mathcal{A}|} \mathcal{L}(\boldsymbol{\theta}, \mathcal{D}_a^y)\|_2},$$

for $\forall a \in \mathcal{A}$. We now consider each element of $\boldsymbol{q}^{y*}$. For given $y \in \mathcal{Y}$, we define $\boldsymbol{\gamma}^y \in \mathbb{R}^{|\mathcal{A}|}$ as follows:

$$\gamma_a^y \triangleq \frac{\mathcal{L}(\boldsymbol{\theta}, \mathcal{D}_a^y) - \sum_a \frac{1}{|\mathcal{A}|} \mathcal{L}(\boldsymbol{\theta}, \mathcal{D}_a^y)}{\|\mathcal{L}(\boldsymbol{\theta}, \mathcal{D}_a^y) - \sum_a \frac{1}{|\mathcal{A}|} \mathcal{L}(\boldsymbol{\theta}, \mathcal{D}_a^y)\|_2},$$

for any $a \in \mathcal{A}$. Fix $i \in \mathcal{A}$ by symmetry. Then, we have:

$$\sum_{a \in \mathcal{A}} \gamma_a^y = 0, \text{ and thus, } \gamma_i^y = - \sum_{a \in \mathcal{A} \backslash i} \gamma_a^y, \tag{A.3}$$

$$\sum_{a \in \mathcal{A}} (\gamma_a^y)^2 = 1, \text{ and thus, } (\gamma_i^y)^2 = 1 - \sum_{a \in \mathcal{A} \backslash i} (\gamma_a^y)^2. \tag{A.4}$$

By the Cauchy-Schwarz inequality,

$$|\gamma_i^y| = |(1, ..., 1) \cdot (\gamma_1^y, ..., \gamma_{i-1}^y, \gamma_{i+1}^y, ..., \gamma_{|\mathcal{A}|}^y)|$$

$$\leq \sqrt{|\mathcal{A}| - 1} \cdot \sqrt{(\gamma_1^y)^2 + \cdots + (\gamma_{i-1}^y)^2 + (\gamma_{i+1}^y)^2 + \cdots + (\gamma_{|\mathcal{A}|}^y)^2}$$

$$= \sqrt{|\mathcal{A}| - 1} \cdot \sqrt{1 - (\gamma_i^y)^2},$$

in which the first and last equality follow from (A.3) and (A.4), respectively.

From the above inequality, we have the followings:

$$(\gamma_i^y)^2 \leq (|\mathcal{A}| - 1) \cdot (1 - (\gamma_i^y)^2),$$

$$|\mathcal{A}|(\gamma_i^y)^2 \leq |\mathcal{A}| - 1,$$

$$|(\gamma_i^y)| \leq \sqrt{\frac{|\mathcal{A}| - 1}{|\mathcal{A}|}}. \tag{A.5}$$

Considering (A.3), the equality in (A.5) is attained if and only if

$$\gamma_i^y = \pm\sqrt{\frac{|\mathcal{A}| - 1}{|\mathcal{A}|}},$$

$$\gamma_a^y = \mp\sqrt{\frac{1}{|\mathcal{A}|(|\mathcal{A}| - 1)}}, \forall a \in \mathcal{A}\backslash i.$$

Thus, we can derive the range described in Proposition 2. ∎

## B  ADDITIONAL RELATED WORKS

### B.1  FAIRNESS-CONSTRAINED OPTIMIZATION

As mentioned in Sec. 2.1, some works (Agarwal et al., 2018; Cotter et al., 2019) employ a constrained optimization framework to enforce the group fairness. They formulate a minimax problem of the Lagrangian function from the given constrained optimization problem and find a saddle point by alternating the outer minimization step w.r.t. model parameters and the inner maximization step w.r.t. the Lagrangian variables. When updating the model parameters, they employ a convex upper bounded surrogate of 0-1 loss such as hinge loss or cross entropy loss in order to address non-differentiability of the fairness constraints. Further, Cotter et al. (2019) employ a non-zero sum formulation that uses the surrogates for updating the model parameters and use the original 0-1 loss for updating the Lagrangian dual variables.

Although both Cotter et al. (2019) and our method similarly solve the minimax problems with 0-1 loss terms by using surrogate losses in the minimization step (w.r.t. model parameters) and the original 0-1 loss in the maximization step, we emphasize that Fair DRO address a different kind of minimax problem compared to that of Cotter et al. (2019). Namely, their approach solves a minimax problem involving both model parameters and Lagrangian variables, while we only focus on the minimization w.r.t. model parameters, with a fixed $\rho$ (which corresponds to a fixed Lagrangian variable). Instead, based on our theoretical results that the minimization problem w.r.t. model parameters can be formulated as another minimax problem based on the DRO framework, FairDRO solves such minimax problem by the non-zero sum formulation.

Furthermore, we argue that our surrogates would be more advantageous in terms of bounding the original objective function. When implementing the algorithm of Cotter et al. (2019), they make surrogates of group fairness constraints by typically using an upper bounded function of 0-1 loss such as the hinge loss or the cross entropy loss. However, the modified fairness constraint terms with such surrogates sometimes are not necessarily sharp upper bounds of the original constraints, *e.g.,* multi-class classification setting. The reason is because the group fairness constraints, such as EO or ECA, are typically defined as the gap of averaged 0-1 losses for measuring the parity over groups. However, if we use the cross entropy loss as a surrogate in equation (6), we can get valid convex upper bound of (7) whenever $q^y$ is a maximizer of the inner maximization problem, and all entries of $q^y$ are positive values. Thus, we expect that our FairDRO can outperform PL by solving the original objective function better, which is shown in our experimental results.

### B.2 Robust optimization in algorithmic fairness

Much more works related to algorithmic fairness have recently utilized the DRO frameworks for their own goals. A line of works embeds the DRO frameworks into the fairness-constraint optimization problem to make a model fair, and robust to distribution shifts of a test dataset (Jiang et al., 2020; Taskesen et al., 2020; Wang et al., 2021; Mandal et al., 2020). For example, Mandal et al. (2020) aimed to train fair classifiers to be with respect to weighted perturbations in the training distribution by considering all possible weight set over the training samples. Wang et al. (2021) and Taskesen et al. (2020) proposed DRO objectives with Wasserstein uncertainty sets which are combined by group fairness constraints. Although all aforementioned methods showed improvements in group fairness and robustness, the key differences from FairDRO are that we leverage the DRO framework for directly promoting group fairness, not robustness, and consider the uncertainty set over groups, not samples.

Some works proposed DRO formulations for achieving fairness in more challenging settings like learning with imperfect group labels or sequential learning. Wang et al. (2020) proposed how to achieve the group fairness successfully given noisy group labels only, by considering the worst-case distribution of group labels. Hashimoto et al. (2018) devised repeated loss minimization using the DRO-based training objective with the $\chi^2$-divergence uncertainty set for encouraging long-term fairness in a sequential setting where the group populations changes over time. However, they do not also use DRO as a direct tool for achieving group fairness.

There are some recent works perhaps closest to our method in terms of using DRO for achieving the fairness. Rezaei et al. (2020) incorporated a group fairness constraint into statistic-matching based robust log-loss minimization problem for training fair logistic regression models. Although they showed nice theoretical benefits in terms of its convexity and convergence, their method has a limitation that group information is needed at the test time. Yurochkin et al. (2019) aims to achieve the fairness of a classifier using DRO frameworks. However, they only focus on a novel *individual* fairness notion called as distributionally robust fairness (DRF), not group fairness.

### B.3 Minimax optimization in algorithmic fairness

Several works adopted min-max optimization techniques for group fairness (although they are not based on DRO frameworks). Zhang et al. (2018) proposed an adversarial debiasing technique that eliminates group information of latent features of a neural network (NN) model through the mini-max game of an adversary and the NN model. Baharlouei et al. (2020) and Jiang et al. (2020) proposed novel regularization terms for group fairness using Rényi correlation and Wasserstein distance respectively, which lead to min-max formulations. We emphasize that these methods rely on additional terms or adversaries for promoting group fairness, while our method directly uses the DRO-based min-max formulation as a fairness regularization term.

## C More implementation details

We used PyTorch (Paszke et al., 2019); Experiments are performed on a server with AMD Ryzen Threadripper PRO 3975WX CPUs and NVIDIA RTX A5000 GPUs. All models are evaluated on separate test sets, and all experiments are repeated with 4 different random seeds. All the reported results are the averaged results.

### C.1 Optimization

For tabular and vision datasets, we train all models with the AdamW optimizer (Loshchilov & Hutter, 2019) for 70 epochs. We set the mini-batch size and the weight decay as 128 and 0.001, respectively. The initial learning rate is set as 0.001 and decayed by cosine annealing technique (Loshchilov & Hutter, 2017).

For the language dataset, we fine-tune pre-trained BERT with the AdamW optimizer for 3 epochs. We set the mini-batch size and the weight decay as 24 and 0.001, respectively. The initial learning rate is set as 0.00002 and adjusted with a learning rate schedule using a warm-up phase followed by a linear decay. All results are reported for the model at the last epoch.

## C.2 IMPLEMENTATION DETAILS OF IN-PROCESSING BASELINES

Here, we describe the implementation details of each in-processing baseline used for the experiments.

The original LBC (Jiang & Nachum, 2020) requires multiple full-training iterations by alternatively re-weighting each group based on the given fairness metric and re-training the full dataset. Since this optimization procedure needs a very high computation budget, we modify full-training iterations to 5 epochs and 14 iterations for vision and language datasets.

FairBatch (Roh et al., 2020) and RW (Kamiran & Calders, 2012) are implemented in the way they were originally proposed.

Cov (Zafar et al., 2017b) utilized a fairness constraint based on Covariance between the group label and the signed distance of the feature vectors from the decision boundary of a classifier. Although they used the Disciplined convex-concave program (DCCP) (Shen et al., 2016) solver for the constrained problem, we apply our optimization procedure (*i.e.*, AdamW) by setting the covariance-based constraint as a regularization term. We note that extenstion Cov to multi-class classifcation tasks is non-trivial because the signed distance is not obviously defined for multi-class decision boundary.

MFD (Jung et al., 2021) and FairHSIC (Quadrianto et al., 2019) use additional fairness-promoting regularization terms based on MMD and HSIC. For FairHSIC, we only implement the second term of their decomposition loss (*i.e.*, the HSIC loss between the feature representations and the group labels). For implementing their regularization terms, we use the Gaussian RBF kernel of which the variance parameter is set as the mean of squared distance between all data points, following Jung et al. (2022). Since regularization terms used in both MFD and FairHSIC are applied to feature vectors of DNN model, they are limited to DNN-based models.

Baharlouei et al. (2020) attempted to remove the non-linear dependency between the model prediction and the sensitive attribute by using Rényi correlation as a regularization term. The resulting training objective is a minimax optimization problem where the maximization is for computing the Rényi correlation, and the minimization is for learning a model. When the group label is binary, we use modified Algorithm 2 in Baharlouei et al. (2020), where the minimization step (*i.e.*, line 4 in Algorithm 2) is processed in a mini-batch manner. Although Algorithm 1 is designed for the non-binary group label, applying the algorithm to a task with non-binary group labels is not straightforward because the second term in the RHS of equation 6 is a biased estimator when using the mini-batch optimization.

## C.3 HYPERPARAMETERS FOR MAIN RESULTS

We perform the grid search on the hyperparameter candidates for every method. The full hyperparameter search space is illustrated in Tab. C.1.

## C.4 MODEL SELECTION RULE FOR A SINGLE EVALUATION METRIC

The accuracy-fairness trade-off may exist in many real-world applications (Dutta et al., 2020); better accuracy leads to worse fairness, and vice versa. For example, Fig. 1 shows the trade-off for the Adult and COMPAS datasets. Therefore, when using a single evaluation the model should be carefully selected for fair evaluation. To that end, we explore varying hyperparameters (HPs) of each method for controlling the strength between accuracy and fairness. We then select the best model showing the best fairness criterion $\Delta_{\mathrm{DCA}}$ while achieving at least 95% of the accuracy of the vanilla-trained model. With this selection rule, we report the best performance of each model throughout Appendix D. The HPs for each method were extensively searched, but fairly in terms of computation budgets. The search range of each hyperparameter is listsed in Tab. C.1.

# D ADDITIONAL RESULTS

## D.1 RESULT TABLES

Following our model selection rule, described in Appendix C.4, we report the best performances of each result in Tab. D.1 and Tab. D.2. The numbers in the parentheses correspond to the standard

Table C.1: **Hyperparameter search spaces.** We perform the grid search to find the best hyperparameters for each method.

| Method | Hyperparameter | Search range |
|---|---|---|
| Cov (Zafar et al., 2017b) | Covariance strength $\lambda$ | $[10^{-2}, 10^2]$ |
| Rényi (Baharlouei et al., 2020) | Rényi correlation strength $\lambda$ | $[10^{-2}, 10^3]$ |
| MFD (Jung et al., 2021) | MMD strength $\lambda$ | $[10^{-2}, 10^4]$ |
| FairHSIC (Quadrianto et al., 2019) | HSIC strength $\lambda$ | $[10^{-2}, 10^4]$ |
| LBC (Jiang & Nachum, 2020) | LR for re-weights $\eta$ | $[10^{-3}, 10^2]$ |
| FairBatch (Roh et al., 2020) | LR for mini-batch size of each group $\alpha$ | $[10^{-3}, 10^2]$ |
| EGR (Agarwal et al., 2018) | LR for Lagrange multiplier $\eta$ 
 $L_1$-norm bound $B$ | $[10^{-2}, 10^2]$ 
 $[10^{-2}, 10^1]$ |
| PL (Cotter et al., 2019) | LR for Lagrange multiplier $\eta$ 
 Violation upper bound $\kappa$ | $[10^{-2}, 10^2]$ 
 $[10^{-2}, 10^1]$ |
| FairDRO | Radius of $\chi^2$-divergence ball $\rho$ | $[10^{-2}, 10^2]$ |

Table D.1: **The best performances on Adult and COMPAS datasets**. The number in the parentheses with $\pm$ stands for the standard deviation of each metric obtained from several independent runs with 4 different seeds. The target accuracy (higher is better) and DCA (1) (lower is better) are shown. We follow the proposed model selection criterion in Appendix C.4.

| | Adult | | COMPAS | |
|---|---|---|---|---|
| | Acc. ($\uparrow$) | $\Delta_{\text{DCA}}$ ($\downarrow$) | Acc. ($\uparrow$) | $\Delta_{\text{DCA}}$ ($\downarrow$) |
| Scratch | 81.95 ($\pm0.03$) | 6.83 ($\pm0.06$) | **63.29** ($\pm0.00$) | 14.51 ($\pm0.00$) |
| RW (Kamiran & Calders, 2012) | **81.98** ($\pm0.04$) | 6.84 ($\pm0.09$) | 63.29 ($\pm0.15$) | 14.87 ($\pm0.62$) |
| LBC (Jiang & Nachum, 2020) | 81.21 ($\pm0.17$) | 4.92 ($\pm0.27$) | 62.04 ($\pm0.24$) | 6.03 ($\pm1.13$) |
| FairBatch (Roh et al., 2020) | 81.63 ($\pm0.03$) | 5.75 ($\pm0.09$) | 62.38 ($\pm0.07$) | 6.71 ($\pm1.07$) |
| Cov (Zafar et al., 2017a) | 78.68 ($\pm0.12$) | 5.08 ($\pm0.51$) | 60.31 ($\pm0.30$) | 7.58 ($\pm3.71$) |
| Rényi (Baharlouei et al., 2020) | 77.09 ($\pm0.21$) | 4.86 ($\pm0.78$) | 61.44 ($\pm0.35$) | 6.77 ($\pm0.86$) |
| EGR (Agarwal et al., 2018) | 80.59 ($\pm0.04$) | 6.03 ($\pm0.07$) | 60.66 ($\pm0.73$) | 7.47 ($\pm0.94$) |
| PL (Cotter et al., 2019) | 79.49 ($\pm0.14$) | 2.55 ($\pm0.39$) | 61.95 ($\pm0.35$) | 6.73 ($\pm0.95$) |
| **FairDRO** | 79.23 ($\pm0.11$) | **1.99** ($\pm0.38$) | 61.97 ($\pm0.24$) | **5.20** ($\pm0.92$) |

deviation. From Tab. D.1 and Tab. D.2, we observe FairDRO achieves the best $\Delta_{\text{DCA}}$ with moderate variances. For example, in Tab. D.1, while the best baselines achieve $\Delta_{\text{DCA}}$ of 2.55 and 6.03 on Adult and COMPAS, respectively, our FairDRO achieves 1.99 and 5.20 for the same datasets, respectively. We also emphasize that although the accuracy of FairDRO is slightly worse than the one of RW and Scratch on COMPAS, FairDRO shows significantly better DCA (5.20) compared to RW (14.87) and Scratch (14.51).

In Tab. D.2, we observe consistent results with Tab. D.1. We again note that we marked results of COV and FairHSIC as N/A because COV is designed only for binary classification and FairHSIC fails to converge due to the gradient exploding. We again observe that FairDRO achieves the best level of the fairness metric (DCA) compared to other methods. We also provide *difference of Equalized Odds* (DEO) on UTKFace, denoted by $\Delta_{\text{DEO}}$, which is defined as follows:

$$\Delta_{\text{DEO}} \triangleq \frac{1}{|\mathcal{Y}|^2} \sum_{y,y' \in \mathcal{Y}} \max_{a,a'} |P(\widehat{Y} = y'|A = a, Y = y) - P(\widehat{Y} = y'|A = a', Y = y)|.$$

Note that $\Delta_{\text{DCA}}$ and $\Delta_{\text{DEO}}$ are the same for binary labels. Although LBC, FairBatch, and FairHSIC were devised for achieving EO, we again show that FairDRO outperforms baselines on UTKFace with respect to DEO.

Table D.2: **The best performances on UTKFace and CivilComments**. The number in the parentheses with $\pm$ stands for the standard deviation of each metric obtained from several independent runs with 4 different seeds. "N/A" denotes the method is not trainable. All other details are the same as Tab. D.1.

|  | UTKFace | | | CivilComments | |
|---|---|---|---|---|---|
|  | Acc. ($\uparrow$) | $\Delta_{DCA}$ ($\downarrow$) | $\Delta_{DEO}$ ($\downarrow$) | Acc. ($\uparrow$) | $\Delta_{DCA}$ ($\downarrow$) |
| Scratch | 78.83 ($\pm$0.35) | 19.42 ($\pm$1.44) | 13.78 ($\pm$0.95) | **78.12** ($\pm$1.28) | 12.24 ($\pm$2.37) |
| RW (Kamiran & Calders, 2012) | 79.27 ($\pm$1.10) | 17.92 ($\pm$0.60) | 11.22 ($\pm$0.00) | 76.58 ($\pm$1.63) | 7.78 ($\pm$1.19) |
| LBC (Jiang & Nachum, 2020) | 75.58 ($\pm$1.84) | 16.92 ($\pm$1.83) | 12.89 ($\pm$1.03) | 75.85 ($\pm$2.17) | **7.04** ($\pm$0.97) |
| FairBatch (Roh et al., 2020) | 78.04 ($\pm$0.76) | 15.92 ($\pm$2.47) | 12.39 ($\pm$0.96) | 77.18 ($\pm$1.30) | 18.91 ($\pm$3.80) |
| Cov (Zafar et al., 2017a) | N/A | | | 76.39 ($\pm$1.11) | 12.18 ($\pm$4.07) |
| FairHSIC (Quadrianto et al., 2019) | 77.35 ($\pm$0.91) | 17.58 ($\pm$1.67) | 12.83 ($\pm$0.73) | N/A | |
| MFD (Jung et al., 2021) | **81.83** ($\pm$0.95) | 16.08 ($\pm$0.76) | 11.08 ($\pm$0.55) | 76.70 ($\pm$1.09) | 11.36 ($\pm$2.78) |
| EGR (Agarwal et al., 2018) | N/A | | | 76.40 ($\pm$3.04) | 9.53 ($\pm$6.22) |
| PL (Cotter et al., 2019) | 78.71 ($\pm$0.67) | 16.42 ($\pm$0.83) | 11.89 ($\pm$0.52) | 78.00 ($\pm$1.67) | 10.38 ($\pm$3.37) |
| **FairDRO** | 79.48 ($\pm$0.59) | **14.25** ($\pm$1.26) | **11.06** ($\pm$1.02) | 77.65 ($\pm$1.75) | 7.07 ($\pm$1.44) |

|  | CelebA | |
|---|---|---|
|  | Acc. ($\uparrow$) | $\Delta_{DCA}$ ($\downarrow$) |
| Scratch | 85.80 ($\pm$0.55) | 14.24 ($\pm$0.84) |
| RW (Kamiran & Calders, 2012) | 77.60 ($\pm$0.68) | 15.49 ($\pm$1.41) |
| LBC (Jiang & Nachum, 2020) | 88.96 ($\pm$0.97) | 6.67 ($\pm$2.39) |
| FairBatch (Roh et al., 2020) | 83.85 ($\pm$0.70) | 10.62 ($\pm$0.86) |
| Cov (Zafar et al., 2017a) | 90.10 ($\pm$0.25) | 7.57 ($\pm$0.84) |
| Rényi (Baharlouei et al., 2020) | 86.22 ($\pm$0.79) | 10.62 ($\pm$1.10) |
| FairHSIC (Quadrianto et al., 2019) | 86.63 ($\pm$0.64) | 9.10 ($\pm$3.19) |
| MFD (Jung et al., 2021) | **90.97** ($\pm$0.35) | 4.44 ($\pm$1.13) |
| PL (Cotter et al., 2019) | 83.23 ($\pm$1.06) | **1.46** ($\pm$0.84) |
| **FairDRO** | 87.95 ($\pm$1.61) | 2.57 ($\pm$0.82) |

Table D.3: **The best performances on CelebA**. Details are the same as Tab. D.1.

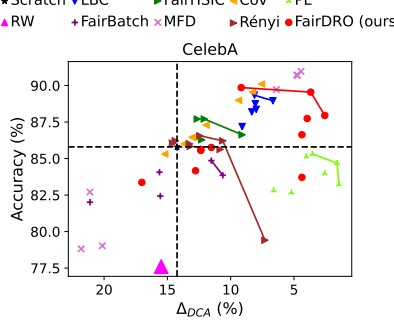

Figure D.1: **Trade-offs between accuracy and DCA on CelebA**. The "Scratch" performances are highlighted by the dotted lines

## D.2 CELEBA RESULTS

To show consistent improvements, we conducted an additional experiment on the CelebA dataset, which includes about 200K celebrities' face images with 40 annotated binary attributes. Among the attributes, we select "blond hair" as the class label and "gender" as the group label, which is the set of attributes widely used in many previous works (Sagawa et al., 2020; Jung et al., 2022; Chuang & Mroueh, 2021; Liu et al., 2021).

Tab. D.3 and Fig. D.1 show a similar result trend to one of UTKFace in Sec. 5.2. FairDRO again outperforms the baselines in terms of DCA and achieves a competitive trade-off with MFD. We also observe re-weighting based baselines show worse DCA than FairDRO, meaning that FairDRO finds the better weights over groups for lower DCA.

## D.3 RESULTS WITH THE STANDARD ERM LOSS

For fair comparison with FairDRO, we used the balanced ERM loss for the training of each baseline as mentioned in Sec. 5. Since all baselines were originally designed for employing the standard ERM, we further reported the results for all methods solving the standard ERM loss. Fig. D.2 show the best performances and the accuracy-fairness trade-offs on Adult, COMPAD, UTKFace, and CivilComments datasets.

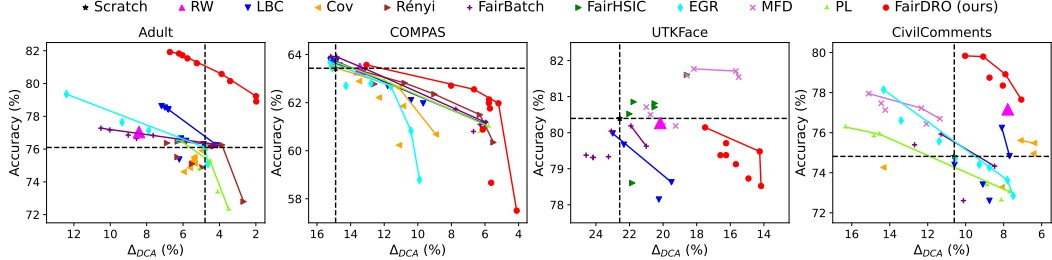

Figure D.2: **The trade-offs between accuracy and DCA**. Each method is trained with the standard ERM. Details are the same as Fig. 1.

Table D.4: **Ablation study of FairDRO**. RVP (w/ classwise) and FairDRO (w/o classwise) are for the ablation study of our key two components, the DRO formulation and classwise treatment, respectively. We follow the same model selection criterion introduced in Appendix C.4.

| | Adult | | COMPAS | | UTKFace | | CivilComments | |
|---|---|---|---|---|---|---|---|---|
| | Acc. ($\uparrow$) | $\Delta_{\mathrm{DCA}}$ ($\downarrow$) | Acc ($\uparrow$) | $\Delta_{\mathrm{DCA}}$ ($\downarrow$) | Acc. ($\uparrow$) | $\Delta_{\mathrm{DCA}}$ ($\downarrow$) | Acc. ($\uparrow$) | $\Delta_{\mathrm{DCA}}$ ($\downarrow$) |
| Scratch | **81.95** | 6.83 | **63.29** | 14.51 | 78.83 | 19.42 | 78.12 | 12.24 |
| FairDRO (w/o DRO) | 80.47 | 4.86 | 62.20 | 8.55 | 79.21 | 17.58 | **78.13** | 9.79 |
| RVP (w/ classwise) | 80.10 | 4.62 | 61.99 | 9.74 | 77.25 | **12.33** | N/A | |
| FairDRO (w/o classwise) | 78.45 | **1.62** | 60.72 | 6.28 | 78.25 | 20.75 | 77.42 | 8.85 |
| **FairDRO** | 79.23 | 1.99 | 61.97 | **5.20** | **79.48** | 14.25 | 77.65 | **7.07** |

## D.4 ABLATION STUDY FOR THE CHOICE OF UNCERTAINTY SET

We continue from Sec. 5.4 and provide further study of FairDRO. We analyze the effect of each constraint employed by FairDRO. Tab. D.5 shows the overall accuracy, $\Delta_{\mathrm{DCA}}$ and the worst accuracy over groups for each ablation case. "Classwise", "$\chi^2$-divergence" and "Quasi-prob" (4) indicate each component added when defining the uncertainty set of Group DRO (*i.e.*, simplex over groups) as described in Sec. 4. For a fair comparison, we apply the same optimization scheme (described in Sec. 4.4) for all DRO variants.

As the supremum in (3) is attained at the vertex of the simplex $\Delta^{|\mathcal{Y}|\times|\mathcal{A}|}$, Group DRO minimizes the worst-case loss over the $(y, a)$ pair, thereby it can prevent the lowest accuracy across the group and class from becoming too low. Thus, Group DRO substantially improves and achieves the best worst-case accuracy over groups compared to ERM (*i.e.*, 1st $\rightarrow$ 2nd row). However, it shows the worst DCA among the DRO variants, confirming the mismatch between the criterion Group DRO uses and the group fairness metric. We observe that changing the simplex uncertainty set of Group DRO to the proposed $\chi^2$-divergence and quasi-prob uncertainty sets (4) (*i.e.*, 2nd $\rightarrow$ 3rd row) has an impact on the group fairness mostly. In addition, reducing the size of uncertainty sets via classwise treatment (*i.e.*, 3rd $\rightarrow$ 6th row) further improves the group fairness. Finally, including negative values in the uncertainty set also helps to improve the group fairness (*i.e.*, 5th $\rightarrow$ 6th row) on most datasets, by more severely penalizing the major groups with high accuracy with negative weights.

Table D.5: **Ablation study for uncertainty set of FairDRO**. We show the effects of each component of the proposed FairDRO: GDRO and the constraints on the uncertainty set, including classwise uncertainty set, $\chi^2$-divergence ball, and quasi-probability. All numbers are average results over Adult, COMPAS, UTKFace, and CivilComments datasets. GDRO is an abbreviation for Group DRO.

| GDRO | Classwise | $\chi^2$-div. | Q-prob. | Adult | | | COMPAS | | | UTKFace | | | CivilComments | | |
|---|---|---|---|---|---|---|---|---|---|---|---|---|---|---|---|
| | | | | Acc. | $\Delta_{\text{DCA}}$ | Worst acc. | Acc. | $\Delta_{\text{DCA}}$ | Worst acc. | Acc. | $\Delta_{\text{DCA}}$ | Worst acc. | Acc. | $\Delta_{\text{DCA}}$ | Worst acc. |
| ✗ | ✗ | ✗ | ✗ | **81.95** | 6.83 | 76.86 | **63.29** | 14.51 | 50.78 | 78.83 | 19.42 | 61.00 | 78.12 | 12.24 | 61.26 |
| ✓ | ✗ | ✗ | ✗ | 81.89 | 6.18 | **78.62** | 62.06 | 6.68 | **57.57** | **79.54** | 18.17 | 61.00 | **80.32** | 12.63 | 69.72 |
| ✓ | ✗ | ✓ | ✓ | 78.45 | **1.62** | 77.46 | 60.72 | 6.28 | 55.94 | 78.25 | 20.75 | 45.75 | 77.42 | 8.85 | 63.70 |
| ✓ | ✓ | ✗ | ✗ | 81.68 | 5.89 | 77.67 | 62.10 | 6.41 | 53.59 | 78.69 | 14.50 | 63.75 | 79.99 | 9.16 | **71.52** |
| ✓ | ✓ | ✓ | ✗ | 81.53 | 5.76 | 77.38 | 62.32 | 6.42 | 53.91 | 78.37 | **13.75** | **66.50** | 78.56 | 7.89 | 67.24 |
| ✓ | ✓ | ✓ | ✓ | 79.23 | 1.99 | 75.60 | 61.97 | **5.20** | 53.91 | 79.48 | 14.25 | 65.75 | 77.65 | **7.07** | 62.85 |

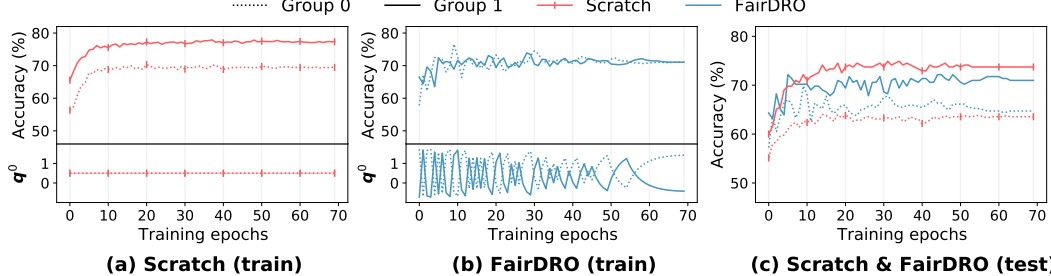

Figure D.3: **Accuracies and loss weights for each group ($q$) on COMPAS** ($y = 0$). Details are the same as Fig. 4.

## D.5 VISUALIZATION OF $q^0$

We continue from Sec. 5.4 and provide the visualization results including the weights ($q^y$), training accuracy, and test accuracy for $y = 0$. In Fig. D.3 (a), Scratch shows that the significant accuracy gap between two groups is kept during all training time. On the other hand, we observe again in Fig. D.3 (b) that $q^0$ initially fluctuates and eventually converges by our optimization procedure, and as a result, the large discrepancy of training accuracy is reduced at the end of training. Finally, we show from Fig. D.3 (c) that the test accuracy gap between the groups of FairDRO is smaller than that of Scratch, *i.e.*, FairDRO achieves lower DCA.

## D.6 ABLATION STUDY OF SMOOTHED IBR

We observe the effectiveness of smoothed IBR updates introduced in Sec. 4.4. Fig. D.4 and Tab. D.6 compares dynamics of $q^1$ and performance on COMPAS depending on whether or not our smoothing technique is used for FairDRO optimization. Jin et al. (2020) showed a theoretical result that the standard IBR asymptotically converge to an approximate stationary point w.r.t. $\theta$ under regularity assumptions. However, from Fig. D.4 (left), we still observe the instability of standard IBR; *i.e.*, $q$ oscillates whenever the group loss fluctuates. On the other hand, Fig. D.4 (right) shows that our smoothing technique has obvious effect on preventing the oscillation of $q$ when using IBR. Furthermore, Tab. D.6 demonstrates that stable dynamics of $q$ can significantly improve performance in terms of DCA.

## D.7 ISSUE OF STANDARD ACCURACY

In the experiments, we reported the balanced accuracy for measuring the performance of the models instead of the standard accuracy. While the balanced accuracy offers several advantages in measuring performance on imbalanced datasets, it also has some drawbacks. In general, it may underestimate the performance of certain groups with relatively large numbers. As in most cases, the groups with large numbers often become majority groups. The balanced accuracy metric may not capture accuracy

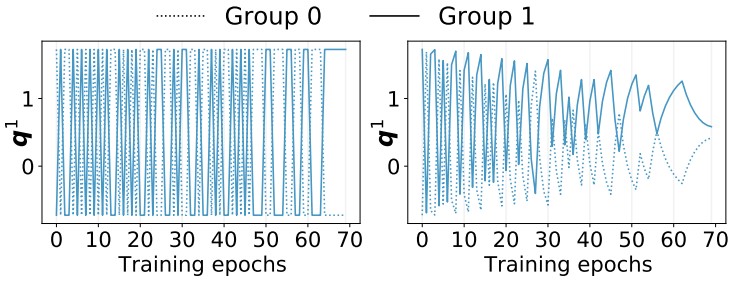

Figure D.4: **$q$ on COMPAS**. The groupwise loss weights during training for label 1 are shown. The reported weights are for FairDRO (w/o smoothing) (left) and FairDRO (right).

Table D.6: **The best performances on Adult and COMPAS datasets**. We follow the same model selection criterion described in Appendix C.4.

|  | Adult | | COMPAS | |
| --- | --- | --- | --- | --- |
|  | Acc. ($\uparrow$) | $\Delta_{\text{DCA}}$ ($\downarrow$) | Acc. ($\uparrow$) | $\Delta_{\text{DCA}}$ ($\downarrow$) |
| Scratch | **81.95** | 6.83 | **63.29** | 14.51 |
| FairDRO (w/o smoothing) | 81.55 | 5.76 | 61.86 | 5.50 |
| **FairDRO** | 79.23 | **1.99** | 61.97 | **5.20** |

Table D.7: **The number of samples in Adult, COMPAS, and CelebA datasets.**

|  | Adult | | COMPAS | | CelebA | |
| --- | --- | --- | --- | --- | --- | --- |
|  | $a = 0$ | $a = 1$ | $a = 0$ | $a = 1$ | $a = 0$ | $a = 1$ |
| $y = 0$ | 13026 | 20988 | 2080 | 1278 | 17809 | 67054 |
| $y = 1$ | 1669 | 9539 | 1987 | 822 | 23060 | 1567 |

Table D.8: **The number of samples in UTKFace and CivilComments datasets.**

|  | UTKFace | | | | CivilComments | | | | |
| --- | --- | --- | --- | --- | --- | --- | --- | --- | --- |
|  | $a = 0$ | $a = 1$ | $a = 2$ | $a = 3$ | $a = 0$ | $a = 1$ | $a = 2$ | $a = 3$ | $a = 4$ |
| $y = 0$ | 2049 | 336 | 1025 | 638 | 11485 | 17632 | 8154 | 4048 | 9823 |
| $y = 1$ | 3735 | 3116 | 1856 | 2221 | 4564 | 5892 | 828 | 545 | 1286 |
| $y = 2$ | 4294 | 1074 | 553 | 1116 | | | - | | |

reduction in majority group. In real-world applications, there is also concern about a decrease in majority-group accuracy.

However, we argue that reporting the vanilla standard accuracy would be problematic when the dataset is severely class-imbalanced. For example, Tab. D.7 shows the number of samples in each dataset with respect to the class and group labels. For the case of Adult dataset, 75.2% of the samples are labeled as $y = 0$, hence, a naive predictor that always predict $\hat{Y} = 0$ will achieve the standard accuracy of 75.2% and DCA= 0, which are clearly overestimating the (accuracy, fairness) performance of the classifier. Therefore, to avoid such issue, we use the group-class balanced accuracy, rather than the standard accuracy.

