# OpenReview forum: "Re-weighting Based Group Fairness Regularization via Classwise Robust Optimization"
_ICLR.cc/2023/Conference — ICLR 2023 poster_

### Official Review · Reviewer_cPS3 · 2022-10-21

**Confidence:** 3
**Correctness:** 3
**Technical Novelty And Significance:** 3
**Empirical Novelty And Significance:** 1
**Recommendation:** 5

**Clarity, Quality, Novelty And Reproducibility:**

Overall the clarity is reasonable, despite many minor typos and grammatical issues. Experiments are described in sufficient detail for reproducibility, but a code release would help a great deal.

**Strength And Weaknesses:**


## Major Comments

* The authors suggest that their approach exactly optimizes for the DCA. However, in practice, they admit to being unable to do so, and instead optimize for a convex relaxation of it by using cross-entropy (Remark, Section 4.3). While this may be a minor point, the entire framing of the paper is based on *exactly solving* the DCA problem, not a relaxation of it. I think the abstract, introduction, and framing throughout should clarify this.

* The experimental results are not particularly good. They seem somewhat promising on 1/2 tabular datasets (Adult) and 1/2 vision datasets (CC), but on the others there is weak, if any, evidence that the proposed method is better. We have no estimates of statistical uncertainty, so it is hard to tell much from the point estimates -- but even more, there is no clear evaluation metric for the prpposed model. Do we care about \delta_DCA? If so,the proposed method doesn't clearly outperform baselines. Do we care about tradeoffs/envelopes? If so, the evidence is mixed at best, with baselines still outperforming the proposed method on at least half the datasets as mentioned. The ablation study is similarly mixed.

	I am particularly concerned about empirical performance because there is already an abundance of theoretically-motivated methods for DRO, as the authors acknowledge in their review, which don't offer much benefit in practice (probably due to being overly pessimistic). If the proposed method doesn't clearly improve, on real tasks according to metrics we as a community agree are important, I don't see a strong justification for publishing yet another robustness method. I'd like to be wrong about this, as I think the authors' direct connection to fairness and accuracy (also an under-investigated metric in the context of robust learning) are important, but the strong theoretical underpinnings should be accompanied by clear empirical results in order to make a case for acceptance.

* It feels somewhat that the authors bury the lede that DCA is equivalent to EO, in the binary case. It would be useful to make this point much earlier, as some of the introduction/setup feels as if it is emphasizing a "new" approach to fair/robustness, but it is really a generalization of a very well-known measure.

## Minor Comments

* It is not intuitively clear to my why we ought to, in principle, balance over classes in ECA (eqn 1). Why is this the right thing to do? It seems to me this could be quite harmful in highly imbalanced datasets, and that this might be part of what is driving the subpar empirical results on many of the experiments.

* The authors should probably cite and discuss the following; it seems relevant to e.g. Sec 4.2: Khani, Fereshte, Aditi Raghunathan, and Percy Liang. "Maximum weighted loss discrepancy." arXiv preprint arXiv:1906.03518 (2019).

* It is often unclear what the authors mean by "directly" optimizing for group fairness (e.g. "both...do not use DRO as a direct tool for achieving group fairness"). Please clarify.

* "Debiasing" is mentioned as an application at least twice; I'm not sure this is an application. I believe "image classification" would be a better description of the application in the work cited as a debiasing paper.

* How is the "arg max" in Equation (11) solved? It is not clear.

* All of the tables are way too small. Please revise.

* Why are the x-axes in Figures 1,2 both reversed?

* The y=0 curve for Fig. 3 are hidden away in Appendix D.4, but those results look qualitatively different -- oscillating wildly between extremes. Please comment on these results.

## Typos etc.

There are *many* grammatical problems with the paper. I list a few here, but it needs a thorough revision by a native English speaker.

Abstract: "between the reweighting-" -> "between reweighting"

P1: "in learning various decision-making" -> "in various decision-making"

P1: "such application is" -> "such applications is"

P1: "re-weighting sample weights" -> "re-weighting samples"

P1: "To address such issue, the fairness-aware" -> "To address such issues, fairness-aware"

P2: "in algorithmic fairness literature" -> "in the..."

P7 "finds the better" -> finds better

**Summary Of The Paper:**

The paper proposes a method for training "fair" models via a classwise variant of Group DRO. The authors show that this is theoretically equivalent to optimizing for Difference of Conditional Accuracy, but they introduce some relaxations to make the problem tractable in practice. Experiments are conducted on two tabular, two image, and two language datasets with comparisons of the proposed method to baselines.


**Summary Of The Review:**

Overall, this is an interesting and promising work, but it feels of borderline quality. It is nice to see DRO taken in the direction of more practical problems grounded directly in the fairness literature. I think that the main contribution of this work is on the theoretical side; on the empirical side, the results are quite mixed, and it is hard to disentangle the effects of some of the many implementation and experimental decisions on the empirical results. There is also not a clear framework for evaluation, which makes it even harder to say whether this method clearly "works" -- or whether it can be added to the pile of theoretically interesting, but practically unhelpful, robust optimization methods. I have other more minor concerns that are detailed above.

---

> ### Author Response · Authors · 2022-11-19
> **Response to reviewer cPS3 (Part 1/2)**
>
> We thank Reviewer cPS3 for positive comments on our theoretical results of connection between DCA and DRO. We also thank for your concerns about our experimental result and our writing in terms of solving DCA, etc. Followings are our reply for your concerns.
>
> **Weakness**
> 1. Thank you for pointing out our misleading statement about solving DCA. What we want to state in this paper is to exactly incorporate the DCA term to our DRO-based objective function, not solving the DCA exactly. However, as you concern, since we optimize for a convex relaxation for DCA, not the exact DCA, we agree that our statements for exactness of the DCA regularization would be misleading. Thus, we clarified this more clearly in the modified title, abstract, introduction, and remark of section 4.3.
> 2. We apologize for the slightly unclear presentations of our experimental results. Before replying to your concern, we note that based on the reviewer QUdb's comment that our evaluation can be unfair for baselines, we again reported the performance of baselines in the revised version by modifying their ERM loss to group-class balanced loss. Since this modification mostly improved the balanced accuracy of baselines while DCA is almost the same, there is no benefit for us. Moreover, we use the fairness-accuracy trade-offs (Figure 1 and 2) only in the main text as the comparison metric (The table results remained in [Table D.1/D.2, Appendix]). Finally, we additionally plotted convex envelopes in the fairness-accuracy trade-offs for visually easier comparison.  Hence, we will mainly reply with the trade-off figures in the revised version.
> We emphasize that from all the original and modified results, FairDRO achieves the best or competitive pareto-trade-offs on all datasets. Although some baselines achieve the best trade-off on certain datasets, their performances are severely worse on different domains of datasets. Specifically, we observe from the modified result that although MFD shows the best trade-offs on vision datasets thanks to knowledge distillation, MFD shows inferior performance on CivilComments to FairDRO, LBC and RW. On the other hand, FairDRO achieves the second best trade-offs on vision datasets and also obtains state-of-the-art best trade-offs on remained three datasets. Thus, FairDRO provides consistently competitive performance on all datasets from various domains, which clearly supports our theoretical results.
> 3. Thank for your comment on emphasizing the equivalence between DCA and EO. We put more emphasis on the equivalence of DCA in our abstract and introduction of the revised version.

---

> > ### Author Response · Authors · 2022-11-19
> > **Response to reviewer cPS3 (Part 2/2)**
> >
> > > Minor questions
> > 1. We thank the reviewer for the comment on the balanced accuracy. This indeed is a philosophical comment, and we argue our reasoning below. ECA is a fairness criterion which requires equal accuracies among groups for each given class. Without balance over classes, it may not be able to capture well the accuracy gap for classes with the small number of data. Further, this phenomenon can be problematic when the dataset is highly class-imbalanced. To that end, we argue that DCA should be balanced across the classes to objectively evaluate the classifier's performance.
> > 2. Thank your for introducing additional related work. Khani et al. [A1] proposed the maximum weighted loss discrepancy (MWLD) loss and showed that MWLD is related to the loss variance and the group fairness varying on a weighting function of MWLD. Further, they showed through experimental results that minimizing the loss variance has an effect on improving group fairness. From these points, [A1] is obviously related to our work. We cited and discussed this paper in [Section B, Appendix] of the revised version.
> > 3. FairDRO utilizes DRO in order to achieve group fairness, but the methods listed do not target group fairness, or adopt DRO for the purpose of robustness to distribution shifts of a test dataset. Additional related works that use DRO framework in the fairness literature are listed in [Section B, Appendix].
> > 4. Thank you for your detailed comments. We also corrected typos reflecting all comments from the reviewers. We will make sure to clarify all the ambiguous or wrong expressions including many grammatical corrections in the final version of the paper.
> > 5. The precise closed-form solution for Eq.(11) is Eq.(8). It can be constructed by considering the equality conditions for the Cauchy-Schwarz inequality with the uncertainty set constraints. The detailed derivation of the maximizer is in the proof of Lemma 1.
> > 6. We thank for your comment. We adjusted the size of all the tables to make the values easier to see. We will make sure to consider other details for clarification in the final version.
> > 7. This is for consistency with accuracy on the $y$-axis. For example, it means that performance improves when accuracy moves away from the origin. Similarly, we inverted the $x$ axis to imply better performance as the DCA moves away from the origin. However, it’s a matter of whether to prefer right-up or left-up, so we can easily change it.
> > 8. We can observe that the fluctuations in both $q^1$ in Fig. 4 and $q^0$ in Figure D.5 decrease as the training continues. At the extremes, each converged smoothly but converged to different points. Unlike $q^1$, $q^0$ converges to the point of assigning a negative value to the group with high accuracy, reducing the training accuracy gap at the end of training, which we believe is somewhat qualitatively similar to $q^1$.
> >
> > **References**
> > - [A1]: Khani et al. Maximum weighted loss discrepancy. Arxiv, 2019.

---

> > > ### Comment · Reviewer_cPS3 · 2022-12-01
> > > **discussion**
> > >
> > > Apologies for the late comment, I did not realize the author discussion phase was ongoing.
> > >
> > > Thank you for the thoughtful replies.
> > >
> > > I am wondering whether the authors can comment on the following:
> > >
> > > * **Baseline results on tabular datasets**: It seems that the accuracy numbers for all models, but particularly the baselines, are lower than expected on Adult and COMPAS. As just one example, in [1] (which is a recent paper, but does not present any new methods/datasets so is still a valid comparison to the current work), we can see in Table 3 that even a simple MLP can achieve 85.7% accuracy on Adult, and 72.4% accuracy on COMPAS (and logistic regression achieves 85.2% and 70.0% accuracy, respectively). I am wondering whether the authors can comment on why all evaluated methods achieve significantly lower accuracy in Figure 1?
> > >
> > > * **Mixed empirical findings**: While the presentation of the results has improved considerably and I acknowledge the authors' efforts on this, it seems that the overall evidence for the proposed method is mixed. I don't believe that a new method needs to strictly dominate all other methods on all datasets, but I am wondering whether the authors are able to provide some intuition for why the method might perform comparatively well on some datasets (i.e. CivilComments) but show much more marginal gains on others. It seems that, if one were to include Clopper-Pearson confidence intervals for accuracy on Figure 1, the proposed method would overlap with the baselines, at most if not all points.
> > >
> > >
> > > [1] Gardner, Josh, Zoran Popović, and Ludwig Schmidt. "Subgroup Robustness Grows On Trees: An Empirical Baseline Investigation." arXiv preprint arXiv:2211.12703 (2022).

---

> > > > ### Author Response · Authors · 2022-12-08
> > > > **Reply for additional comments**
> > > >
> > > > Thank you very much for your comments. The followings are our follow-ups.
> > > > * Please note that we reported the **group-class balanced accuracy**, not the standard accuracy, because the datasets are severely imbalanced over groups and classes (please refer to Table D.10 and Section D.7 for more details). When the accuracy of a group with a fewer number of data points is severely lower than other groups as in many typical fairness problems, the balanced accuracy can be lower than the standard accuracy. For example, in our settings, the standard accuracies of our Scratch model (which is simply trained with the ordinary ERM loss as in our original submitted version) on Adult and COMPAS are 84.56% and 65.86%, respectively, while the balanced accuracies are 76.10% and 63.48%, respectively (as shown in Table D.4). Moreover, we note that one of the additional reasons for our method achieving lower accuracy on COMPAS than [A2] is that we use less data points. Namely, following the pre-processing of [A1], we filtered out 1047 incomplete data points and used 6167 remained data points, while [A2] used 7214 data points. Additionally, we note that the balanced accuracy we reported on Adult and COMPAS, 76.1% and 63.48%, is nearly the same as results in [A1], 75% and 65%, (The slight differences would be stemming from the optimization details, e.g., batch size, optimizer, etc.).
> > > > * We provided experimental results on three domains of datasets, _i.e._, tabular (Adult and COMPAS), Vision (UTKFace), and NLP (CivilComments). Assuming that the level of difficulty for achieving better accuracy-fairness tradeoff for the tabular datasets is relatively lower than those of Vision and NLP datasets, we can infer from our results that the performance gain of our method widens for more complex and challenging datasets. Namely, for relatively simple datasets such as Adult and COMPAS, other baselines based on less exact surrogates (such as PL, Cov) or heuristics (such as LBC, FairBatch) can achieve comparable accuracy-fairness trade-offs to our method. On the other hand, for more complex datasets, UTKFace and CivilComments, our method achieves comparatively better trade-offs than such baselines with more exact DCA regularization.(We observe that FairHSIC and MFD achieve better accuracy on UTKFace due to the regularization effect for accuracy, but such effect does not occur on the NLP dataset, leading to poor trade-offs.) We further emphasize that for tabular datasets, our FairDRO can provide pareto-frontiers in a wider range than other re-weighting baselines by introducing negative weights.
> > > > - [A1] : Rachel K. E. Bellamy. AI Fairness 360: An Extensible Toolkit for Detecting, Understanding, and Mitigating Unwanted Algorithmic Bias, IBM Journal of Research and Development, 2019.
> > > > - [A2] : Gardner et al. Subgroup Robustness Grows On Trees: An Empirical Baseline Investigation., ArXiv, 2022.

---

> > > ### Comment · Area_Chair_MBH5 · 2022-12-07
> > > **Re: discussion**
> > >
> > > Authors: if you have any thoughts on the questions from the last message of **Reviewer cPS3**, please let us know.

---

### Official Review · Reviewer_U6MV · 2022-10-22

**Confidence:** 3
**Correctness:** 4
**Technical Novelty And Significance:** 3
**Empirical Novelty And Significance:** 3
**Recommendation:** 6

**Clarity, Quality, Novelty And Reproducibility:**

I think the connection between DCA and DRO is interesting. Even though both the objectives seek to enforce equitable performance across groups, establishing a precise connection is a nice contribution. The specific algorithm used in the experiments is not new.

Where the paper lacks is in discussion of how similar approaches to handling non-differentiability already exist in the group fairness literature (e.g. Cotter et al., Agarwal et al.), and a comparison with principled constrained optimization methods that more directly seek to enforce DCA.



**Strength And Weaknesses:**

Pros:
- The connection between enforcing DCA (which is typically imposed as constraints in the optimization problem) and DRO is pretty interesting.
- The experimental results are elaborate enough to show case that DRO can indeed be used to achieve competitive DCA performance

Cons:
- The paper seems to emphasize that the prescribed use of DRO allows one to tackle the non-differentiability of DCA. I think as the authors themselves point out, even after the DRO reformulation, the objective is still intractable, and one does have to use surrogates for at least one step of the min-max optimization. In fact, similar "selective of use of surrogates" is common even in the constrained optimization literature. For example, the approach of Cotter et al (2019) formulates a constrained optimization problem to enforce constraints such as equalized odds, and does so by using surrogates for only updates on the model parameters and uses the original 0-1 loss for the group weights. So in terms of being able to address the non-differentiability of DCA, I don't this paper offering a solution that is very different what already exists in the literature.

- A important baseline that is missing is the use of constrained optimization in the above manner to enforce DCA. I see that authors include Zafar et al. (2017b), but this method uses covariance-based surrogates, which I doubt are as tight as the more common cross-entropy style losses. I think either Cotter et al. (2019), which uses surrogates in the same way as this paper, but does so through a Lagrangian formulation instead of DRO, is one compelling baseline to include. Agarwal et al (2018) is another compelling baseline, which comes with theoretical guarantees --  this method uses a Lagrangian formulation based on the 0-1 loss to generate example weights, and solves the resulting cost-weighted optimization problem using a standard cost-sensitive learning (where one could employ a surrogate of choice). From what I can tell, the techniques in both Cotter et al. and Agarwal et al. are amenable to metrics like DCA.

Ref:
Cotter et al., "Optimization with Non-Differentiable Constraints with Applications to Fairness, Recall, Churn, and Other Goals", JMLR 2019.

Minor questions:
- In the experiments, is there a reason why the "Scratch" method, which if I understand correctly seeks to optimize a classification loss without fairness constraints, doesn't always do the best on accuracy (e.g. Adult, UTK faces, Civil comments)?
- In discussing prior re-weighting methods, you mention "they lack sound theoretical justifications for enforcing group fairness". I don't think this is particularly true of methods like Agarwal et al.


**Summary Of The Paper:**

The paper seeks to enforce the group fairness metric "Difference of  Conditional Accuracy", a generalization of the equalized odds criterion, and points out an interesting connection between optimizing this metric and (class-wise) distributionally robust optimization (DRO). Specifically, they show that the criterion can be approximately written in terms of variances of group losses, and uses a result from Xie et al. (2010) to show an equivalence to DRO. Experiments on a number of benchmark datasets show case that the use class-wise DRO can indeed achieve better trade-offs between accuracy and DCA.

**Summary Of The Review:**

The paper makes an interesting connection between DCA and DRO, but requires a more elaborate discussion on some related methods and comparing against them in the experiments.

---

> ### Author Response · Authors · 2022-11-19
> **Response to reviewer U6MV**
>
> We thank Reviewer U6MV for positive comments,  e.g., connection between DCA and DRO is interesting and empirical results are elaborate. We also thank for pointing out missing discussions about important related works. Followings are our reply for your concerns.
>
> **Weakness**
> 1. We sincerely apologize for missing discussions about differences from Cotter et al. [A1] and thank you very much for pointing out. We first note that although their approach and our method similarly solve minimax problems with 0-1 loss terms by using surrogates in minimization steps (w.r.t. model parameters) and original 0-1 loss in maximization steps, we address a different kind of minimax problem from Cotter et al. As you mentioned, the approach of Cotter et al. solves the minimax problem of Lagrangian formulation by using surrogates of 0-1 loss constraints for updates of model parameters. On the other hand, we only focus on a minimization w.r.t. model parameters, given the $\rho$ which corresponds to a Lagrangian variable. Thus, based on our theoretical results that the minimization problem w.r.t. model parameters can be formulated as another minimax problem based on the DRO framework, FairDRO solves such minimax problem by the selective use of surrogate for model parameters.
> Secondly, we construct surrogates for our objective function itself by changing 0-1 loss to cross entropy loss in our GroupDRO formulation, while Cotter et al. use surrogates for only 0-1 loss constraints. We argue that our surrogates would be more advantageous in terms of better bounding the original objective function. When implementing the algorithm of Cotter et al., they make surrogates of group fairness constraints by typically using an upper bounded function of 0-1 loss such as the hinge loss or the cross entropy loss. However, the modified fairness constraint term with such surrogates is not an upper bound of the original constraints because a group fairness constraint such as equalized odds is typically defined as the gap of averaged 0-1 losses for measuring a parity over groups. However, if we use the cross entropy loss as a surrogate in  Eq.(6), we can get the convex upper of Eq.(7) whenever $q$ is a maximizer of the inner maximization problem and all entries of $q$ are positive values. Thus, we expect that our FairDRO can provide better solutions in the minimization step of model parameters, which is supported by our experimental results in the following reply.
> Furthermore, we expect that our FairDRO can be used as a better plug-in method for updates of model parameters in the algorithm of Cotter et al., which will be an interesting future work.
>
>
> 2. Thank you for pointing out the missing baselines. The following table compares Expontiated Gradient Reduction (EGR) [A2] and Proxy Lagrangian (PL) [A1] on two tabular datasets (The accuracy-fairness trade-offs of the two methods are also plotted in the revised version). We observe again that FairDRO outperforms EGR and PL in DCA and show better accuracy-fairness trade-offs. We can infer that this performance gap is due to the way the surrogates are used, as we mentioned above. We will report results on other vision and NLP datasets in the final version.
>
>     |           | Adult |                | COMPAS |                |
>     | --------- |:-----:|:--------------:|:------:|:--------------:|
>     |           | Acc.  | $\Delta_{DCA}$ |  Acc.  | $\Delta_{DCA}$ |
>     | Scratch   | 80.87 |      15.58     | 63.12  |     16.56      |
>     | EGR       | 81.73 |      7.17      | 60.66  |      7.47      |
>     | PL        | 78.86 |      2.62      | 62.19  |      5.93      |
>     | FairDRO   | 79.23 |      1.99      | 61.97  |      5.20      |
>
> > Minor questions
>
> 1. [**Same as the third response to QUdb**] Since "Scratch" in the original version employs the standard ERM, the balanced accuracy of Scratch can be lower than FairDRO or other baselines when the class and group of a dataset are highly imbalanced. In the revised version, the accuracies of Scratch are competitive or better, compared to the accuracy FairDRO. Please check [Table D.1/D.2, Appendix] in the revised version.
> 2. We apologize for our wrong descriptions for prior re-weighting methods. We modified ``they`` to ``most of them`` in the sentence you pointed out and added a discussion about Agarwal et al. [A2] in Section 2.1 and Appendix of the revised version.
>
> **References**
> - [A1]: Cotter et al. Optimization with Non-Differentiable Constraints with Applications to Fairness, Recall, Churn, and Other Goals. JMLR, 2019.
> - [A2]: Agarwal et al. A Reductions Approach to Fair Classification. ICML, 2018.

---

### Official Review · Reviewer_QUdb · 2022-10-24

**Confidence:** 3
**Correctness:** 2
**Technical Novelty And Significance:** 3
**Empirical Novelty And Significance:** 2
**Recommendation:** 6

**Clarity, Quality, Novelty And Reproducibility:**

The authors reveal a tight connection between the standard deviation of the losses and the DCA. This result, particularly the lower bound part, is non-trivial. Combining with Xie et al.'s result is an interesting idea, and the resulting algorithm is reasonable and theoretically grounded.

I found a similarity between the present algorithm and reduction-based fair classification approaches, including
- Agarwal et al. A Reductions Approach to Fair Classification. In ICML 2018.
- Cotter et al. Optimization with Non-Differentiable Constraints with Applications to Fairness, Recall, Churn, and Other Goals. JMLR, 2019.
These algorithms also carry out alternating updates consisting of the update of the model parameter and the update of the group-wise scalar parameters. The authors should clarify the essential difference between the proposed and these algorithms.

I have some concerns about the empirical evaluations.
1. The authors state that the accuracy is measured using balanced classification accuracy. Using such a measure as assessing accuracy is no problem. However, the existing methods, including Scratch, might employ the standard (not balanced) loss as their objective function. It is unfair that only the proposed method minimizes the balanced loss directly, whereas the other methods might not.
2. In Figures 1 and 2 (and related figures in the appendix), there are cases where FairDRO achieves higher accuracy than Scratch. It is very weird because Scratch optimizes the accuracy without a constraint of fairness. The authors should clarify why such phenomena happen.
3. The authors report the minimum unfairness criterion while achieving at least 95% of the original accuracy. However, comparing this measure is not a good way because the accuracy is not aligned. It is better to compare the Pareto frontier between the comparison methods.
4. The ablation study does not adequately demonstrate the necessity of each component. In Table 3, RVP achieves the best DCA in UTKFace, and FairDRO(w/o classwise) reaches the best DCA in Adult. Hence, it fails to adequately demonstrate the optimal performance of FairDRO, as claimed in the main body.


Minor comments:
- The convergence of Algorithm 1 is crucial property. It is better to clarify what assumptions are needed for convergence and discuss the satisfiability of these assumptions in standard situations.
- The statement about $C$ in Corollary 1 is somewhat misleading. The paper says $C$ is some positive constant, which looks like there is a constant $C$ independent of the dataset. However, $C$ might be changed by the dataset, indeed. An accurate statement would be that for any dataset $D$, there is a corresponding constant $C$ such that theta achieves Eq. 7. It is better to revise the statement so that it does not mislead.

**Strength And Weaknesses:**

Strength:
- Clearly written and easy to follow
- A tight connection between DCA and variance of losses is interesting and non-trivial.
- The proposed algorithm is novel and reasonable.

Weakness:
- A lack of comparison with the reduction-based methods.
- The empirical evaluations have some concerns.


**Summary Of The Paper:**

The authors investigate the classification problem with the penalization of the difference in conditional accuracy. They provide a tight characterization between the empirical DCA and variance of 0-1 losses. By combining it with the result of Xie et al., they derive the group-wise distributionally robust optimization form, which is more tractable than the original form. They also provide an iterative algorithm for solving the group DRO form of their optimization by utilizing the closed-form solution of the group DRO part. The empirical comparison demonstrates that the proposed algorithm is competitive or outperforms the existing fair classification method for equal opportunity. Also, the ablation study shows the robustness of the proposed mechanisms.

**Summary Of The Review:**

This paper is well-written and easy to follow. The tight characterization between DCA and the standard deviation of the losses is interesting and significant. The developed algorithm is novel and reasonable. Hence, I recommend acceptance at this point. However, the paper lacks a comparison with the crucial related work. I also have a concern about the empirical evaluations. I'd like the authors to address these concerns.

---

> ### Author Response · Authors · 2022-11-19
> **Response to reviewer QUdb (Part 1/2)**
>
> We thank Reviewer QUdb for positive comments,  e.g., the non-trivial connection between DCA and variance of losses, interesting combination of the DRO formulation and reasonable resulting algorithm. We also thank for pointing out missing related works and our empirical evaluations. Followings are our reply for your concerns.
>
> > I found a similarity between the present algorithm and reduction-based fair classification approaches.
> * Thank you for introducing two related works. Although both their work and our FairDRO similarly utilize an alternative updating scheme for solving a minimax problem as the reviewer mentioned, arguments in the inner maximization problem of the related works and FairDRO are essentially different. The two related works employ a constrained optimization framework to enforce group fairness. They formulate a minimax problem of the Lagrangian function from the given constrained optimization problem and find a saddle point by alternating the outer minimization step w.r.t. model parameters and the inner maximization step w.r.t. the Lagrangian variables. On the other hand, FairDRO is a regularization-based method whose objective function is the right side of Eq.(7). Namely, the radius $\rho$ which is corresponding to the Lagrangian variable above is a tunable hyperparameter, not a learnable parameter (It does not necessarily mean that solving a constrainted problem is more efficient in terms of the number of hyperparameters because a constrained problem needs additional hyperparameter for controlling the degree of constraints and a learning rate of Lagrangian variable). Given the $\rho$, FairDRO minimizes its objective function by alternatively updating model parameters and distributions over groups. To sum up, FairDRO focuses on how the minimization step w.r.t model parameters proceeds effectively given Lagrangian variable, while the prior two works consider how we find the optimum primal and Lagrangian variable of the constrained optimization.
>
> > I have some concerns about the empirical evaluations
> 1. We agree with the reviewer's comments on the unfairness in our evaluations. To address this issue, we re-evaluated baselines including "Scratch" by modifying the standard loss to the group-class balanced loss so that their objective functions are balanced over each group-class pair as in ours. (We did not re-evaluate MFD since MFD uses the group-class balanced batch sampler for stable approximation of MMD loss, i.e., MFD already employs the balanced loss.) We revised the experimental section of our manuscript with the modified results in Figure 1 and Figure 2. From the renewed results in the revised paper, we observe that while the balanced accuracies of baselines are mostly improved, our FairDRO still consistently shows competitive or the best fairness-accuracy trade-offs on all datasets.
> 2. Since "Scratch" in the original version employed the _standard_ ERM, the balanced accuracy of Scratch can be lower than FairDRO or other baselines when the classes and groups of a dataset are highly imbalanced. In [Table D.1 and D.2, Appendix] of the revised version, the (balanced) accuracies of Scratch are competitive or better, compared to the accuracy of FairDRO and other baselines.
> 3. Thank you for the valid comment. The main reasoning behind our evaluation criterion was due to the hardness of exactly aligning the accuracies of all comparing baselines. However, we agree that simply showing the average accuracy/DCA numbers could cause some misunderstanding, hence, we instead reported the full accuracy-DCA trade-off plots in Section 5 of the revision. Moreover, in Figure 1 and Figure 2, we additionally plotted the convex envelope of the trade-off curves for varying hyperparameters, i.e., the pareto-frontier for a clearer comparison among different methods.
> 4. In response to the comment, we also reported the pareto-frontiers for our ablation study [Figure 3, Section 5], instead of the best DCA. From the pareto-frontier figures on four datasets, we clearly observe that FairDRO achieves competitive or best trade-offs, compared to RVP and FairDRO (w/o classwise). Thus, we can derive a conclusion that two components of FairDRO, the classwise treatment and DRO formulation, are indeed needed for the best performance.

---

> > ### Author Response · Authors · 2022-11-19
> > **Response to reviewer QUdb (Part 2/2)**
> >
> > > Minor questions
> >
> > 5. [**Same as the second reply to uGcU**] There exist theoretical results [A1] for the convergence guarantee of IBR algorithm. In [A1], they show that a nonconvex-concave minimax problem can be solved by IBR algorithm (which is called GDmax in [A1]) with a convergence guarantee (refer to Theorem E.3 [A1]) under the mild assumptions of smoothness of the objective function. Although our smoothed IBR algorithm is not equipped with a convergence guarantee, there are some recent works [A2, A3] showing that a smoothing scheme similar to ours can stabilize the oscillation during optimization of minimax problems and ensure convergence to a stationary solution. We indeed observed through experimental results that the smoothing technique has an obvious effect on preventing oscillation of $q$ from Figure 4, Figure D.5, and Figure D.6, leading to better performances than the standard IBR as shown in [Table D.8, Appendix]. We will add this discussion in the final version.
> > 6. We agree to the reviewer's comment on the unclear part of the statement in our Corollary 1. We modified ``for some constant $C$`` to ``a corresponding constant $C_{D}$ to given dataset $D$`` in the revised version.
> >
> > **References**
> > - [A1]: Lin et al. On Gradient Descent for Nonconvex-Concave Minimax Problems. ICML, 2020.
> > - [A2]: Tatjana et al. Taming Gans with Lookahead-Minmax. ICLR, 2021.
> > - [A3]: Jiawei et al. A Single-Loop Smoothed Gradient Descent-Ascent Algorithm for Nonconvex-Concave Min-Max Problems. NeurIPS, 2020.

---

### Official Review · Reviewer_uGcU · 2022-10-25

**Confidence:** 4
**Correctness:** 4
**Technical Novelty And Significance:** 2
**Empirical Novelty And Significance:** 3
**Recommendation:** 6

**Clarity, Quality, Novelty And Reproducibility:**

Clarity: the paper is well-written, and the connection with prior work is highlighted.

Quality and Novelty: I doubt if the proposed approach is applicable beyond the fairness measure DCA. In terms of novelty, the authors make the connection between DCA and root of empirical variance. However, the solution of the regularized ERM uses the group DRO formulation shown in [Xie et. al. 2020]. This is the only step where distributionally robust optimization comes into the picture, and I wouldn't call such an application of DRO well motivated.

**Strength And Weaknesses:**

Strengths:
- The connection between DCA and empirical variance is interesting and I wonder if one can show similar connections for other fairness measures.
- The experimental results clearly show that the proposed method pareto dominates several other in-processing based methods.

Weaknesses:
- The proposed approach seems limited to difference of conditional accuracy (DCA) and it is not clear how this approach can be generalized to other fairness metrics. In fact, the main algorithm is motivated by proposition 1 which shows that empirical DCA can be approximated by the square root of empirical variance.
- The authors claim that the iterative method for solving fair-DRO is efficient. However, I didn't see any convergence guarantees for the proposed method.
- The authors also emphasized the use of exact regularization in the learning objective. However, this approach cannot guarantee any desired level of fairness in the training/test set. In particular, if one wants to output a classifier with fairness violation at most $\alpha$, the current approach seems to require a lot of hyperparameter tuning.
- There is no result on generalization i.e. how does the fairness guarantee and accuracy generalizes to unseen datasets.


**Summary Of The Paper:**

This paper proposes an in-processing based method to design a fair classifier for the metric difference of conditional accuracy (DCA). First, the authors show that empirical DCA can be approximated by the root of empirical group variance. This suggests using the root of empirical group variance as a regularizer in the learning objective. The authors then use a characterization due to [Xie et. al., 2020] to convert such a regularized ERM objective to a group distributional robust optimization.

The authors provide an iterative method to solve the fair DRO problem where the min player $\theta$ uses gradient descent and the max player $q$ uses smoothed best response. This algorithm is evaluated on several datasets (tabular, vision, and language) and compared with several reweighing and regularization based methods. The results show that compared to the existing fair classifiers, the proposed method achieves better trade-off between accuracy and DCA.

**Summary Of The Review:**

Overall, the authors present an interesting approach to design a fair classifier and the experimental results demonstrate that the proposed method pareto dominates existing fair classifiers on several datasets. However, I wonder if the proposed approach works beyond the fairness metric DCA. Moreover, the main connection with DRO relies on past literature [Xie et. al. 2020] and the proposed method lacks convergence and generalization guarantees.

Update after rebuttal: Many thanks to the authors for the detailed response. It seems to me that the connection between fair classification and DRO is indeed non-trivial, as pointed out by other reviewers. Additionally, the new experimental results look convincing to me. Furthermore, the fairness metric DCA covers the more interesting fairness metrics for binary classification. But I hope the authors add some discussion on how generalizable this approach is to other fairness measures, particularly for multi-class classification. Overall I now feel positive about the paper and will increase my score.

---

> ### Author Response · Authors · 2022-11-19
> **Response to reviewer uGcU**
>
> We thank Reviewer uGcU for positive comments, e.g., the connection between DCA and our method is interesting, and our experimental results clearly show the effectiveness of our method. We also thank for the comments about our limitations and lack of convergence and generalization guarantees. Followings are our follow-up on your concerns.
>
> **Weakness**
> 1. Thank you for pointing out our limitation. Although our method considers one fairness notion, ECA, it is the generalization of two popular group fairness metrics, **Equalized Odds (EO)** and **Equal Opportunity (EOpp)**, as we discussed on the page 3. In the binary class label setting, ECA is equivalent to EO because both criteria require equal TPR and TNR. Furthermore, we argue that ECA is a natural extension of EOpp to the multi-class setting because ECA requires equality of TPR among groups, which resembles the argument of EOpp. Furthermore, our method can be simply generalized to another popular group fairness notion, **equal accuracy (EA)**, by constructing a group DRO formulation without classwise treatment. We will add a discussion and experiment results for the extension to EA in the final version.
> 2. We apologize for missing discussions about the convergence guarantee of our optimization algorithm. There already exist theoretical results [A1] for the convergence guarantee of IBR algorithm. In [A1], they show that a nonconvex-concave minimax problem can be solved by IBR algorithm (which is called GDmax in [A1]) with a convergence guarantee (refer to Theorem E.3 [A1]) under the mild assumptions of smoothness of the objective function. Although our smoothed IBR algorithm is not equipped with a convergence guarantee, there are some recent works [A2, A3] showing that a smoothing scheme similar to ours can stabilize the oscillation during optimization of minimax problems and ensure convergence to a stationary solution. We indeed observed through experimental results that the smoothing technique has an obvious effect on preventing oscillation of $q$ from Figure 4, Figure D.5, and Figure D.6, leading to better performances than the standard IBR as shown in [Table D.9, Appendix]. We will add this discussion in the final version.
> 3. All regularization-based methods including ours and baselines require a hyperparameter tuning for achieving the desired level of fairness. Employing a constrained optimization problem such as Cotter et al. [A4] is a solution for controlling the degree of fairness on the training set, but it requires tuning additional hyperparameters such as learning rate for the Lagrangian variable and burdens higher computational cost for finding an optimal point. Further, solving the constrained optimization problem to achieve a certain degree of fairness on the training set does not always generalize to the test set, which means that it also needs an additional hyperparameter tuning for achieving satisfactory performance on the test set. Thus, we argue that such hyperparameter tuning for achieving a certain level of fairness is inevitable in existing fair training methods, and our FairDRO can provide better trade-offs than other fair training methods under the same burden for hyperparameter tuning as shown in our experimental results [Table C.1, Appendix].
> 4. There exist theoretical results for the train-test generalization guarantee, shown in [Theorem 1, A5]. It can be applied to our case when the solution obtained by FairDRO is a minimizer of DCA-constrained optimization problem. We will discuss the generalization of our method in the final version.
>
> **Clarity, Quality, Novelty and Reproducibility**
> > In terms of novelty, the authors make the connection between DCA and root of empirical variance. However, the solution of the regularized ERM uses the group DRO formulation shown in [Xie et. al. 2020]. This is the only step where distributionally robust optimization comes into the picture, and I wouldn't call such an application of DRO well motivated.
> * As Reviewers QUdb, U6MV and cPS3 have also said, we believe making a clear connection between the DCA and DRO is non-trivial and significant. Moreover, thanks to the DRO formulation of the DCA-regularized ERM problem, solving the non-differentiable DCA regularization is converted to a more tractable minimax optimization, which sufficiently motivates the use of DRO framework to enforce DCA.
>
> **References**
> - [A1]: Lin et al. On Gradient Descent for Nonconvex-Concave Minimax Problems. ICML, 2020.
> - [A2]: Tatjana et al. Taming Gans with Lookahead-Minmax. ICLR, 2021.
> - [A3]: Jiawei et al. A Single-Loop Smoothed Gradient Descent-Ascent Algorithm for Nonconvex-Concave Min-Max Problems. NeurIPS, 2020.
> - [A4]: Cotter et al. Optimization with Non-Differentiable Constraints with Applications to Fairness, Recall, Churn, and Other Goals. JMLR, 2019.
> - [A5]: Donini et al. Empirical Risk Minimization Under Fairness Constraints. Neurips, 2018.

---

### Author Response · Authors · 2022-11-19
**General Response and Summary of Updates to Manuscript**

We appreciate all the reviewers for giving your time and all the constructive comments. We submitted a revised manuscript and would like to summarize the changes we made in the revision. Followings are the main revised contents:
* **[Experimental results]** Considering the feedback by [QUdb] and the questions by [U6MV, cPS3], we revised several parts in the experimental sections. For fairer evaluations, we changed the main results by modifying the objective functions of baselines such that loss for each target class is balanced over each class-group label pair. For a clearer evaluation protocol, we reported only the accuracy-fairness trade-offs in the main text and moved the table results to Appendix D. We also plotted convex envelopes of the trade-offs points for visually easier comparisons in Figure 1 and Figure 2. We hope that these modified results can be more persuasive and convey clearer messages to the reviewers.
* **[Related works]**  We apologize for missing out some important related works (as [QUdb, U6MV, cPS3] have mentioned). In Section 2, we listed the additional related works that employ a constrained optimization problem to enforce fairness and show the similar connection between fairness and variance. We also argued how they are different from our FairDRO.
* **[Exact regularization of DCA]** Considering the feedback by [cPS3], we clarified our statements about exact regularization and the use of a convex relaxation in our optimization algorithm. To this end, the title, abstract, introduction and remark in Section 4.3 are revised.
* **[Other missing parts & minor corrections]** We corrected the statement of Corollary 1 [QUdb] and some typos [cPS3].

---

### Decision · Program_Chairs · 2023-01-20

**Decision:**

Accept: poster

**Justification For Why Not Higher Score:**

Empirical results could be more extensive, as noted by at least two reviewers.

**Justification For Why Not Lower Score:**

Interesting new theoretical framework, as appreciated by some domain expert reviewers.

**Metareview: Summary, Strengths And Weaknesses:**

The paper proposes a new method for exactly optimizing the difference-of-conditional-accuracy fairness metric, by casting the problem as a class-wise DRO objective. An efficient optimization scheme is provided for the resulting min-max objective.

Reviewers generally found the main insight that relates fairness constraints to a form of DRO to be interesting, and novel. One concern was regarding the broad applicability of the result (i.e., whether it was overly specialized to a few fairness metrics). The response and reviewer discussion noted that these metrics are canonical in the fairness literature. I agree with this view, and find the results of the paper interesting.

There was more concern on the empirical results, which appear to feature somewhat lower numbers than prior work, and originally missed some natural baselines (and discussion thereof). The response included some new results and discussion of prior methods. There were still some questions about the lower numbers compared to earlier works. The resolution of these would make a much stronger case for the paper.

Without these, the central issue is whether the theoretical findings are of sufficient interest in their own right. Given the favorable response from domain experts, I lean towards accepting the paper, with the strong encouragement of clearly spelling out the many points that arose in the response, and clarifying the scope of the empirical results.

**Note From Pc:**

if the above contains the word "oral" or "spotlight" please see: "oral" presentation means -> notable-top-5% and "spotlight" means -> notable-top-25%. As stated in our emails, we are disassociating presentation type from AC recommendations

**Summary Of Ac-Reviewer Meeting:**

See below summary. Reviewers were generally in agreement that the paper has some interesting theoretical contributions, though it could have some scope for improvement in the empirical side. The reviewers critical of the latter did not see them as prohibitive for the paper eventually being accepted.

_Attendees_

AC, Reviewers uGcU, U6MV, cPS3, QUdb.

_Meeting notes_

Reviewer uGcU: main concern was around the scope of results: it seemed highly specialised to a few fairness measures. It is not clear whether it generalises. The response is somewhat clarifying, but still leaning on the side of rejection.

Reviewer U6MV: approached the paper more from the theoretical perspective. The measures they consider, like EO, are standard in the fairness literature. The theoretical connection they draw between a non-differentiable fairness constraint and a min-max approach are interesting. One critique was that the treatment of existing DRO methods was a bit limited. The response has clarified that somewhat.

Reviewer cPS3: main concerns were around the empirical results. In particular, some of the numbers reported in Table 1 are much lower than what is achievable in the literature. In some settings, it is not clear that the proposed methods offer significant gains. Also, the treatment of existing DRO methods is not extensive.

Reviewer QUdb: theoretical results are interesting. Agreed with the points raised by cPS3 about the experiments. However, overall the author response appears satisfactory.

Reviewer U6MV: agreed with the points raised by cPS3 about the experiments having scope for improvement. One comment is that the lower numbers could be the result of using a weaker model class.

Reviewer uGcU: agreed that if one could argue that the measures are indeed the most relevant ones for fairness, the assessment of the significance could be improved.

Reviewer cPS3: would not argue that the paper be rejected solely on the basis of the experimental results; if the theory is seen as strong enough, then there could be a case for acceptance.